# Semantic Feature Learning for Universal Unsupervised Cross-Domain Retrieval

**Lixu Wang**
Northwestern University, IL, USA
`lixuwang2025@u.northwestern.edu`

**Xinyu Du**
General Motors Global R&D, MI,USA
`xinyu.du@gm.com`

**Qi Zhu**
Northwestern University, IL, USA
`qzhu@northwestern.edu`

## Abstract

Cross-domain retrieval (CDR) is finding increasingly broad applications across various domains. However, existing efforts have several major limitations, with the most critical being their reliance on accurate supervision. Recent studies thus focus on achieving unsupervised CDR, but they typically assume that the category spaces across domains are identical, an assumption that is often unrealistic in real-world scenarios. This is because only through dedicated and comprehensive analysis can the category composition of a data domain be obtained, which contradicts the premise of unsupervised scenarios. Therefore, in this work, we introduce the problem of Underline{U}niversal Underline{U}nsupervised Underline{C}ross-Underline{D}omain Underline{R}etrieval ($\text{U}^2\text{CDR}$) for the first time and design a two-stage semantic feature learning framework to address it. In the first stage, a cross-domain unified prototypical structure is established under the guidance of an instance-prototype-mixed contrastive loss and a semantic-enhanced loss, to counteract category space differences. In the second stage, through a modified adversarial training mechanism, we ensure minimal changes for the established prototypical structure during domain alignment, enabling more accurate nearest-neighbor searching. Extensive experiments across multiple datasets and scenarios, including close-set, partial, and open-set CDR, demonstrate that our approach significantly outperforms existing state-of-the-art CDR methods and other related methods in solving $\text{U}^2\text{CDR}$ challenges.

## 1 Introduction

In real-world applications, cross-domain retrieval (CDR) finds extensive utility across diverse domains, such as image search [1], product recommendations [2], and artistic creation [3, 4]. However, the efficacy of current CDR methods relies heavily on accurate and sufficient supervision [5, 6] to provide categorical or cross-domain pairing labels. The acquisition of such information demands costly efforts and resources. Hence, there is an urgent need to develop unsupervised CDR techniques.

For the regular Unsupervised CDR (UCDR) problem [7, 8], there are two data domains with semantic similarity but distinct characteristics: the query domain and the retrieval domain. Despite the absence of category labels, regular UCDR typically assumes that the label spaces of both domains are identical. However, in real-world applications [5], ***the categorical composition of an unlabeled data domain is usually uncertain, which is hard to acquire without detailed analysis and dedicated expertise***. In this work, we focus on extending UCDR to more universal scenarios, which allow for the possibility of disparate category spaces across domains. The objective of this ***Universal UCDR*** ($\text{U}^2\text{CDR}$) problem is to retrieve samples from the retrieval domain that share the same category label with a query sample

38th Conference on Neural Information Processing Systems (NeurIPS 2024).

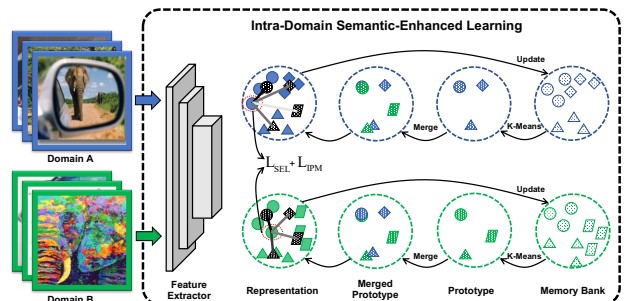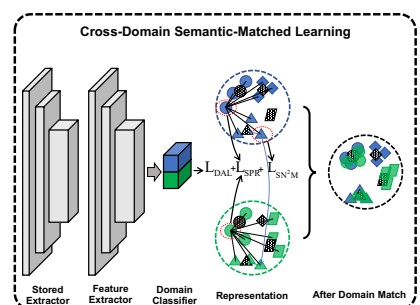

Figure 1: Overview of our proposed UEM semantic feature learning framework for U$^2$CDR. In the first stage of Intra-Domain Semantic-Enhanced Learning, UEM establishes a unified prototypical structure across domains, which is driven and enhanced by an instance-prototype-mixed contrastive loss and a semantic-enhanced loss. In the second stage of Cross-Domain Semantic-Matched Learning, Semantic-Preserving Domain Alignment aligns domains while preserving the built prototypical structure, and Switchable Nearest Neighboring Match achieves more accurate cross-domain categorical pairing.

from the query domain. Naturally, in the case of private categories exclusive to the query domain, the retrieval result should be null.

Two issues must be addressed in solving traditional UCDR: 1) effectively distinguishing data samples in each domain, and 2) achieving alignment across domains for samples of the same category. For the first issue, self-supervised learning [9, 10] (SSL) is employed independently within each domain. For the second, many nearest-neighbor searching algorithms may apply. However, for U$^2$CDR, applying these existing methods introduces new challenges. First, the prevailing SSL methods, particularly contrastive learning [9, 11, 10], are highly influenced by the category label space [12], which means different label spaces lead to distinct semantic structures. Second, existing nearest-neighbor searching algorithms [13, 14, 7] overlook the presence of domain gaps. We found that only by first addressing the domain gap can the nearest neighbor searching become reliable and accurate.

Thus, to effectively address the above challenges in solving U$^2$CDR, we propose a two-stage Unified, Enhanced, and Matched (UEM) semantic feature learning framework, as in Figure 1. In the first stage, we establish a cross-domain unified prototypical structure with an instance-prototype-mixed (IPM) contrastive loss, accompanied by a semantic-enhanced loss (SEL). In the second stage, before conducting cross-domain category alignment, we incorporate Semantic-Preserving Domain Alignment (SPDA) to diminish the domain gap while ensuring minimal changes for the established prototypical structure. As the domain gap diminishes, we propose Switchable Nearest Neighboring Match (SN$^2$M), to select more reliable cross-domain neighbors based on the relationship between instances and prototypes. Extensive experiments and ablation studies on popular benchmark datasets demonstrate that our method can substantially outperform state-of-the-art methods from UCDR and other related problems. In addition, we also theoretically analyze the principles behind the major challenges of U$^2$CDR and the design intuition of UEM. In summary, this work made the following contributions:

- We are the first to identify and solve an important problem when employing UCDR in practice – Universal UCDR (U$^2$CDR), where the category spaces of different domains are distinct.

- We propose a two-stage Unified, Enhanced, and Matched (UEM) semantic feature learning framework to solve U$^2$CDR. In the first stage, UEM establishes a unified prototypical structure across domains, to ensure consistent semantic learning under category space differences. In the second stage, UEM achieves more effective domain alignment and cross-domain pairing.

- We conduct extensive experiments on multiple benchmark datasets, with settings including Close-set, Partial, and Open-set UCDR. The results demonstrate that UEM can substantially outperform state-of-the-art methods of UCDR and other potential solutions in all settings.

## 2 Related Work

**Cross-Domain Retrieval** (CDR) is not very difficult to achieve if there are categorical labels [15, 16]. However, in real-world applications, such categorical labeling information is hard to acquire, thus

more recent works [7, 8, 17] focus on achieving unsupervised CDR (UCDR). CDS [14] proposes a contrastive learning-based cross-domain pre-training to align different domains. PCS [13] incorporates prototype contrastive learning [11] into the cross-domain pre-training. Recent studies also search ways like clustering [7], pseudo-labeling [18], classifier mixup [17], and data augmentation [8] to achieve more advanced CDR. However, all these UCDR works assume that the query and retrieval domains share the same category space. Although there is a study [19] that can achieve CDR with distinct categories, its effectiveness relies on accurate and sufficient data labels.

**Universal Cross-Domain Learning.** Cross-domain learning consists of domain adaptation (DA) and domain generalization (DG) [20]. Regular DA and DG also only consider the scenario where the label space of the target domain is the same as the source label space, which is termed Close-set DA/DG. Recently, more studies have realized that the target label space may be a subset of the source one (i.e., Partial DA/DG) [21, 22] or contain some private labels that other domains do not have (i.e., Open-set DA/DG) [23, 24]. To deal with more universal setups, UniDA [25] unifies entropy and domain similarity to quantify sample transferability across domains. CMU [26] extends transferability quantification into entropy, consistency, and confidence. More recent works search ways like clustering [27, 28] and nearest neighbor matching [29, 30] to achieve universal DA/DG. In addition, some studies appear to achieve unsupervised DG where the source domain is also unlabeled [31, 32], which is similar to the setup of UCDR. However, these studies consider the classification task and cannot effectively work in image retrieval, especially in completely unsupervised cases.

## 3 Methodology

### 3.1 Problem Formulation

In the problem of $\text{U}^2\text{CDR}$, we assume there are two domains characterized by $N^\text{A}$ and $N^\text{B}$ unlabeled instances, which are denoted as $\mathcal{D}^\text{A} = \{\boldsymbol{x}_i^\text{A}\}_{i=1}^{N^\text{A}}$ and $\mathcal{D}^\text{B} = \{\boldsymbol{x}_i^\text{B}\}_{i=1}^{N^\text{B}}$, respectively. Although these two domains are provided as unlabeled data without category labels, we assume their label spaces $\mathcal{Y}^\text{A}, \mathcal{Y}^\text{B}$ consist of $C^\text{A}$ and $C^\text{B}$ different categories, and there is a relationship that $C^\text{A} \neq C^\text{B}, \mathcal{Y}^\text{A} \cap \mathcal{Y}^\text{B} \neq \mathcal{Y}^\text{A} \cup \mathcal{Y}^\text{B}$. Without losing generality, if we regard domain A as the query domain, while domain B is the retrieval domain, the objective of $\text{U}^2\text{CDR}$ is to retrieve correct data from domain B that belongs to the same categories as the query data provided by domain A. To achieve this objective, it is required to train a valid feature extractor $f_\theta : \mathcal{X} \to \mathcal{R}$ that can map both these two domains from the input space $\mathcal{X}$ to a feature space $\mathcal{R}$. Then the retrieval process $R(f_\theta, \boldsymbol{x}_i^\text{A})$ is shaped like for a particular query instance $\boldsymbol{x}_i^\text{A}$ with the label $y_i^\text{A}$ from domain A, the representation distance between all instances in domain B and $\boldsymbol{x}_i^\text{A}$ needs to be calculated to form a set, i.e., $\mathcal{S} = \{\text{d}(f(\boldsymbol{x}_j^\text{B}), f(\boldsymbol{x}_i^\text{A}))\}_{j=1}^{N^\text{B}}$ where $\text{d}(\cdot)$ is a particular distance metric (e.g., Euclidean Distance), and we have

$$R(f_\theta, \boldsymbol{x}_i^\text{A}) = \begin{cases} \text{null, if } y_i^\text{A} \in \mathcal{Y}^\text{A} \setminus \mathcal{Y}^\text{B} \\ \text{sort}_\uparrow(\mathcal{S})[1:k], \text{ otherwise,} \end{cases} \tag{1}$$

where $\text{sort}_\uparrow(\cdot)$ means ascending order sorting, and $[1:k]$ denotes the first $k$ elements of a set.

**Method Overview.** To solve $\text{U}^2\text{CDR}$, we propose a Unified, Enhanced, and Matched (UEM) semantic feature learning framework that consists of two stages – Intra-Domain Semantic-Enhanced (IDSE, Section 3.2) Learning and Cross-Domain Semantic-Matched (CDSM, Section 3.3) Learning, which is shown in Figure 1. IDSE can help the feature extractor $f_\theta$ to extract categorical semantics and ensure a unified semantic structure across domains at the same time, which is achieved by instance-prototype-mixed (IPM, Section 3.2.1) contrastive learning and a novel Semantic-Enhanced Loss (SEL, Section 3.2.2). After IDSE, CDSM conducts Semantic-Preserving Domain Alignment (SPDA, Section 3.3.1) to minimize the domain gap while preserving the semantic structure learned by IDSE. With the minimization of the domain gap, more accurate nearest-neighbor searching can be achieved by our Switchable Nearest Neighboring Match (SN$^2$M, Section 3.3.2).

### 3.2 Intra-Domain Semantic-Enhanced Learning

To achieve effective cross-domain retrieval, feature extractor $f_\theta$ needs to learn consistent cross-domain features to differentiate data categories. Instance Discrimination [9] is usually employed to

achieve discriminative feature learning, but directly applying it in $\mathtt{U}^2\mathtt{CDR}$ has four fundamental issues that hinder the possibility of accurate cross-domain categorical matching later:

1) Instance discrimination tends to extract semantics that separate domains rather than categories,

$$\mathrm{d}(f(\boldsymbol{x}_i^{\mathrm{A}}), f(\boldsymbol{x}_j^{\mathrm{A}})) < \mathrm{d}(f(\boldsymbol{x}_i^{\mathrm{A}}), f(\boldsymbol{x}_j^{\mathrm{B}})),\, y_i^{\mathrm{A}} = y_j^{\mathrm{B}} \neq y_j^{\mathrm{A}}. \tag{2}$$

2) Instance discrimination cannot characterize categorical semantics in the feature space,

$$\frac{\mathrm{d}(\boldsymbol{x}_i^{\mathrm{A}}, \boldsymbol{x}_{j_1}^{\mathrm{A}})}{\mathrm{d}(\boldsymbol{x}_i^{\mathrm{A}}, \boldsymbol{x}_{j_2}^{\mathrm{A}})} < \frac{\mathrm{d}(f(\boldsymbol{x}_i^{\mathrm{A}}), f(\boldsymbol{x}_{j_1}^{\mathrm{A}}))}{\mathrm{d}(f(\boldsymbol{x}_i^{\mathrm{A}}), f(\boldsymbol{x}_{j_2}^{\mathrm{A}}))},\, y_i^{\mathrm{A}} = y_{j_1}^{\mathrm{A}} \neq y_{j_2}^{\mathrm{A}}. \tag{3}$$

3) The randomness introduced by stochastic data augmentations results in evident changes in learned categorical semantic structures during training, i.e.,

$$\mathrm{d}(G(\mathcal{P}_t^{\mathrm{A}}), G(\mathcal{P}_{t+1}^{\mathrm{A}})) \gg \min_{\mathcal{P}_i^{\mathrm{A}}, \mathcal{P}_j^{\mathrm{A}} \sim \mathcal{H}} \mathrm{d}(G(\mathcal{P}_i^{\mathrm{A}}), G(\mathcal{P}_j^{\mathrm{A}})), \tag{4}$$

where $G(\cdot)$ corresponds to a graph constructed by the input vectors, and $\mathcal{P}$ denotes the set of categorical prototypes for a domain. The subscript $t$ denotes different training iterations, while $\mathcal{H}$ represents the hypothesis space of possible categorical prototype sets for a particular domain. $\mathrm{d}(\cdot)$ here is a measurement for graph difference, e.g., graph edit distance [33].

4) Distinct label spaces make instance discrimination learn different categorical semantic structures:

**Theorem 3.1** (Geometry Distinctness). *Suppose data distributions of two domains (A and B) have mutually disjoint supports, and they are uniform over these supports. Assuming the support sets of domains A and B are not identical, the optimal feature extractors $f^*$ that minimize the instance discrimination loss of different domains present distinct geometric feature spaces.*

### 3.2.1 Instance-Prototype-Mixed Contrastive Learning.

To fix the above issues, we adopt a slowly momentum-updated contrastive learning algorithm – MoCo [10], to handle the third issue reflected by Eq. (4). Moreover, the MoCo-based instance discrimination is conducted separately for each domain, which encourages $f_\theta$ to focus less on learning domain semantics (for the first issue, Eq. (2)). Besides, we accompany MoCo with a prototypical contrastive loss, to enhance the mapping of categorical semantics from the input space to the feature space (for addressing the second issue, Eq. (3)). With a well-crafted prototype update mechanism, this prototypical contrastive loss can also help build a unified semantic structure across domains (for the last issue, Theorem 3.1). Then let us introduce $\mathtt{IPM}$ contrastive learning in detail. First of all, two memory banks $\mathcal{M}^{\mathrm{A}}$ and $\mathcal{M}^{\mathrm{B}}$ are maintained for domains A and B, which store historical features $\boldsymbol{m}$ of data samples $\boldsymbol{x}$:

$$\mathcal{M}^{\mathrm{A}} = \left[\boldsymbol{m}_1^{\mathrm{A}}, ..., \boldsymbol{m}_{N^{\mathrm{A}}}^{\mathrm{A}}\right], \mathcal{M}^{\mathrm{B}} = \left[\boldsymbol{m}_1^{\mathrm{B}}, ..., \mathrm{m}_{N^{\mathrm{B}}}^{\mathrm{B}}\right], \text{where } \boldsymbol{m}_i \leftarrow \beta \boldsymbol{m}_i + (1 - \beta) f_\theta(\boldsymbol{x}_i). \tag{5}$$

Here $\boldsymbol{m}_i$ is initialized by the feature of $\boldsymbol{x}_i$ extracted by the initial $f_\theta$ and updated in momentum, where $\beta$ controls the momentum speed, and we set it as a popular value $0.99$. With these two memory banks, MoCo builds the positive pairs as the pair of each instance and its historical feature, while the negative ones are pairs of each instance and the historical features of all other instances:

$$\mathcal{L}_{\mathrm{INCE}} = \sum_{i=1}^{B} -\log \frac{\exp\left(f_\theta(\boldsymbol{x}_i) \cdot \boldsymbol{m}_i / \tau\right)}{\sum_{j=1}^{B} \exp\left(f_\theta(\boldsymbol{x}_i) \cdot \boldsymbol{m}_j / \tau\right)}, \tag{6}$$

where $B$ is the batch size and $\tau$ is a temperature factor that is set as $0.07$.

As for the design of our prototypical contrastive loss, K-Means is applied on $\mathcal{M}^{\mathrm{A}}$ and $\mathcal{M}^{\mathrm{B}}$ to construct prototypes as cluster centers $\mathcal{P} = \{\boldsymbol{p}_c\}_{c=1}^{\widehat{C}}$. In our problem, the cluster number $C$ is unknown, thus we apply the Elbow approach [34] to estimate it as $\widehat{C}$. Then, for each instance $\boldsymbol{x}_i$, if it belongs to the $c_i$-th cluster, the prototypical contrastive loss $\mathcal{L}_{\mathrm{PNCE}}$ shapes like,

$$\mathcal{L}_{\mathrm{PNCE}} = \sum_{i=1}^{B} -\log \frac{\exp\left(f_\theta(\boldsymbol{x}_i) \cdot \boldsymbol{p}_{c_i} / \tau\right)}{\sum_{c=1}^{\widehat{C}} \exp\left(f_\theta(\boldsymbol{x}_i) \cdot \boldsymbol{p}_c / \tau\right)}. \tag{7}$$

Until here, the first three issues can be fixed by mixing $\mathcal{L}_{\mathrm{INCE}}$ and $\mathcal{L}_{\mathrm{PNCE}}$, but the last issue, Theorem 3.1, is the bottleneck of $\mathtt{U}^2\mathtt{CDR}$. Next, let us introduce how we build a unified prototypical

structure for $\mathcal{L}_{\mathrm{PNCE}}$ to address the last issue. Specifically, after obtaining the prototype sets $\mathcal{P}^{\mathrm{A}}, \mathcal{P}^{\mathrm{B}}$ of domain A and B, if we take domain A as an example to illustrate the prototypical structure building process, the prototype set $\mathcal{P}^{\mathrm{B}}$ of domain B will be translated to domain A as,

$$\mathcal{P}^{\mathrm{B}\to\mathrm{A}} = \{\boldsymbol{p}_c^{\mathrm{B}\to\mathrm{A}} = \overrightarrow{\boldsymbol{p}_c^{\mathrm{B}}} + \overrightarrow{\overline{\mathcal{M}^{\mathrm{B}}}\,\overline{\mathcal{M}^{\mathrm{A}}}}\}_{c=1}^{\widehat{C}^{\mathrm{B}}}, \tag{8}$$

where $\overline{\mathcal{M}}$ denotes the average vector of all vectors in $\mathcal{M}$. Next, each prototype $\boldsymbol{p}_c^{\mathrm{A}} \in \mathcal{P}^{\mathrm{A}}$ searches its closest $\boldsymbol{p}_{c'}^{\mathrm{B}\to\mathrm{A}} \in \mathcal{P}^{\mathrm{B}\to\mathrm{A}}$ (we use Hungarian algorithm to search the closest cross-domain prototypes) for the opportunity of merging, which needs to satisfy the condition,

$$d(\boldsymbol{p}_{c'}^{\mathrm{B}\to\mathrm{A}}, \boldsymbol{p}_c^{\mathrm{A}}) < \min\left[\min_{\boldsymbol{p}_i, \boldsymbol{p}_j \in \mathcal{P}^{\mathrm{A}}} d(\boldsymbol{p}_i, \boldsymbol{p}_j), \min_{\boldsymbol{p}_i, \boldsymbol{p}_j \in \mathcal{P}^{\mathrm{B}}} d(\boldsymbol{p}_i, \boldsymbol{p}_j)\right], \tag{9}$$

where $d(\cdot, \cdot)$ computes the Euclidean distance. If we use the symbol $\oplus$ to identify prototypes that satisfy this merging condition, the final prototypical structure for domain A is $\mathcal{P}^{\mathrm{A}'} = \left(\mathcal{P}^{\mathrm{A}} \setminus \mathcal{P}^{\mathrm{A},\oplus}\right) \cup \left(\mathcal{P}^{\mathrm{B}\to\mathrm{A}} \setminus \mathcal{P}^{\mathrm{B}\to\mathrm{A},\oplus}\right) \cup \left(\mathcal{P}^{\mathrm{A},\oplus} \oplus \mathcal{P}^{\mathrm{B}\to\mathrm{A},\oplus}\right)$ where $\left(\mathcal{P}^{\mathrm{A},\oplus} \oplus \mathcal{P}^{\mathrm{B}\to\mathrm{A},\oplus}\right) = \left\{\left(\boldsymbol{p}_{c'}^{\mathrm{B}\to\mathrm{A},\oplus} + \boldsymbol{p}_c^{\mathrm{A},\oplus}\right)/2\right\}_{c=1}^{C^{\oplus}}$. Then the computation of $\mathcal{L}_{\mathrm{PNCE}}^{\mathrm{A}'}$ – Eq. (7) for domain A is conducted on the newly-built $\mathcal{P}^{\mathrm{A}'}$, and all these operations are same for domain B.

However, as establishing the semantic cluster-like structure requires time, it is unreasonable to conduct prototype contrastive learning from the beginning of training. Therefore, we conduct instance discrimination at the beginning and progressively incorporate $\mathcal{L}_{\mathrm{PNCE}}$. In this case, not only are the constructed cluster centers more reliable, but the Elbow approach also provides more accurate cluster number estimations. Specifically, we use a coefficient $\alpha$ that is scheduled by a Sigmoid function to control the incorporation weight of $\mathcal{L}_{\mathrm{PNCE}}$, i.e.,

$$\mathcal{L}_{\mathrm{IPM}} = \mathcal{L}_{\mathrm{INCE}} + \alpha\mathcal{L}_{\mathrm{PNCE}}, \text{ where } \alpha = \frac{1}{1 + \exp(0.5E - e)}, \tag{10}$$

$E$ and $e$ here are the overall training epochs and the current epoch for IDSE.

### 3.2.2 Semantic-Enhanced Loss.

For the IPM contrastive learning, it is arbitrary to allocate a data instance $\boldsymbol{x}_i$ to a single cluster when the preferred semantic prototypical structure cannot be learned in advance. As a result, to speed up the structure-building process, we propose a novel *Semantic-Enhanced Loss* (SEL) to align data instances with the prototypes better. Specifically, instead of assigning data instances with a single cluster, SEL considers potential semantic relationships between instances with all clusters, which are measured by the Softmax probability. Moreover, as both $\mathcal{P}^{\mathrm{A}}$ and $\mathcal{P}^{\mathrm{B}}$ are obtained by the Euclidean distance-based K-Means, we directly minimize the Euclidean distance between samples and prototypes:

$$\mathcal{L}_{\mathrm{SEL}} = \frac{1}{B}\sum_{i=1}^{B}\sum_{c=1}^{\widetilde{C}} \frac{\exp(f_\theta(\boldsymbol{x}_i) \cdot \boldsymbol{p}_c/\tau)}{\sum\limits_{c=1}^{\widetilde{C}} \exp(f_\theta(\boldsymbol{x}_i) \cdot \boldsymbol{p}_c/\tau)} d(f_\theta(\boldsymbol{x}_i), \boldsymbol{p}_c), \tag{11}$$

where $\widetilde{C}$ denotes the number of prototypes after merging, e.g., $\widetilde{C}^{\mathrm{A}}$ is the number of elements in $\mathcal{P}^{\mathrm{A}'}$. By taking all potential semantic correlations into account, SEL can alleviate the impact of the noise within the K-Means clustering results and further guide the model to learn more distinguishable semantic information in terms of Euclidean distance. Certainly, such SEL benefits also rely on high-quality semantic prototypical structures. As a result, we also apply the progressive coefficient $\alpha$ to SEL, then the final optimization objective for IDSE is

$$\mathcal{L}_{\mathrm{IDSE}} = (\mathcal{L}_{\mathrm{IPM}}^{\mathrm{A}} + \mathcal{L}_{\mathrm{IPM}}^{\mathrm{B}}) + \alpha(\mathcal{L}_{\mathrm{SEL}}^{\mathrm{A}} + \mathcal{L}_{\mathrm{SEL}}^{\mathrm{B}}). \tag{12}$$

## 3.3 Cross-Domain Semantic-Matched Learning

### 3.3.1 Semantic-Preserving Domain Alignment.

Domain invariance is another requirement for the extracted features in UEM. However, it is difficult to effectively align feature clusters across domains when no category label nor correspondence

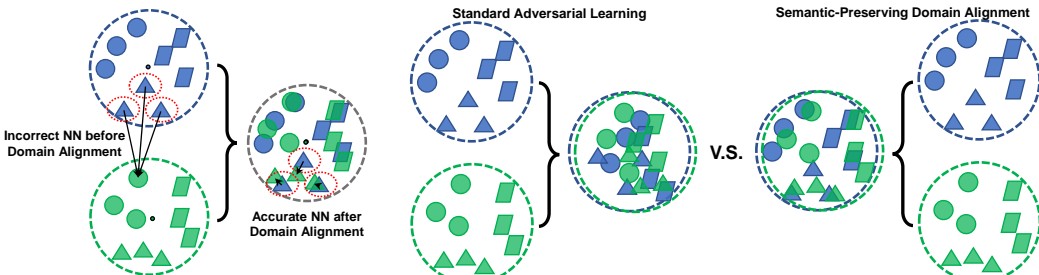

Figure 2: Comparison of nearest neighbor searching before or after domain alignment.

Figure 3: Comparison between standard adversarial learning and our semantic-preserving domain alignment in terms of semantic structure changes.

annotation can be utilized in $\mathtt{U^2CDR}$. Instance matching [7, 32, 13] is proposed to match an instance $\boldsymbol{x}_i^{\mathrm{A}}$ to another instance $\boldsymbol{x}_j^{\mathrm{B}}$ in the other domain with the most similar features. However, due to the domain gap, instances can be easily mapped mistakenly. For example, if there is an instance in one domain that is extremely close to the other domain, it will be determined as the nearest neighbor for all instances in the other domain [13], shown in Figure 2. As a result, before conducting instance matching, the domain gap needs to be diminished. Existing works usually leverage discrepancy minimization [35] or adversarial learning [36] to achieve domain alignment. However, these methods provide inferior performance due to semantic categorical structure changes, i.e., the semantic correlations among instances within domains change a great deal during domain alignment, shown in Figure 3.

To achieve more effective domain alignment, especially with semantic preservation, we propose Semantic-Preserving Domain Alignment (SPDA). Similar to the standard domain adversarial learning, SPDA happens on two parties, one is the feature extractor $f_\theta$, and the other is a domain classifier $g_\omega$ parameterized on $\omega$. The domain classifier tries to distinguish the representations of two domains, while the feature extractor tries to fool the domain classifier. Thus, the training is shaped like a bi-level optimization in terms of the domain classification task on $\theta$ and $\omega$, shown as follows,

$$\mathcal{L}_{\mathrm{DAL}} = \sum_{i=1}^{2B} -y_i \cdot \log g_\omega(f_\theta(\boldsymbol{x}_i)) - (1 - y_i) \cdot \log(1 - g_\omega(f_\theta(\boldsymbol{x}_i))), \tag{13}$$

where $y$ here denotes the domain label, e.g., if we regard $y = 1$ for domain A, the domain label of domain B is $y = 0$. After sufficient adversarial training, the feature extractor captures nearly domain-invariant features, thus achieving domain alignment.

To prevent the semantic structure from being changed, we first make a copy for the model trained by $\mathtt{IDSE}$ and denote it as $f'_\theta$. Then for a mini-batch of a particular domain, we feed all instances to $f'_\theta$ and calculate pair-wise cosine similarity and Euclidean distance. In this way, all these instances constrain and influence each other, which means that when the correlation of a particular instance pair changes, it will affect the correlations of other related pairs. Then, we apply a semantic-preserving regulation into domain adversarial learning to make the pair-wise correlations unchanged,

$$\mathcal{L}_{\mathrm{SPR}} = \frac{1}{B^2} \sum_{i=1}^{B} \sum_{j=1}^{B} \left\{ \left[ \frac{f_\theta(\boldsymbol{x}_i) \cdot f_\theta(\boldsymbol{x}_j)}{|f_\theta(\boldsymbol{x}_i)||f_\theta(\boldsymbol{x}_j)|} - \frac{f_{\theta'}(\boldsymbol{x}_i) \cdot f_{\theta'}(\boldsymbol{x}_j)}{|f_{\theta'}(\boldsymbol{x}_i)||f_{\theta'}(\boldsymbol{x}_j)|} \right]^2 \right.$$
$$\left. + [d(f_\theta(\boldsymbol{x}_i), f_\theta(\boldsymbol{x}_j)) - d(f_{\theta'}(\boldsymbol{x}_i), f_{\theta'}(\boldsymbol{x}_j))]^2 \right\}. \tag{14}$$

**Recall $\mathtt{IPM}$ Contrastive Learning.** Actually, aligning two domains together without any semantic structure change is impossible. As a result, there is a need for a dedicated design to alleviate the impact of such unavoidable changes. Our solution is to strengthen the instances' semantic correlations by enhancing the cluster's inner density and inter-separability. For the final convergence of $\mathtt{IPM}$ contrastive learning, each instance is optimized to get as close to its corresponding cluster prototype as possible, and as far to other cluster prototypes as possible:

**Theorem 3.2** (Convergence of $\mathtt{IPM}$). *Suppose the data distribution of a domain has mutually disjoint supports, and it is uniform over these supports. Simplex Equiangular Tight Frame (ETF) representations [37] minimize the Instance-Prototype-Mixed Loss of this domain.*

### 3.3.2 Switchable Nearest Neighboring Match.

With SPDA, the domain gap could be effectively minimized to enable more accurate cross-domain instance matching. However, existing instance matching approaches [7, 13, 32] lack the capability of measuring the matching reliability between an instance and its nearest neighbor, which allows us to conduct cross-domain matching with different weights. For example, if an instance is located at the joint boundary of multiple categories, which indicates that the current feature extractor cannot extract sufficiently distinguishable semantic features for this instance, we are supposed to lay less emphasis on this case. To fix such issues, we propose the Switchable Nearest Neighboring Match (SN$^2$M).

The principle behind SN$^2$M is that prototypes are more convincing and reliable. Specifically, for a particular sample $x_i^{\mathrm{A}}$ in domain A, we first determine its inner nearest cluster prototype $p_{c_i}^{\mathrm{A}}$ in domain A. We can also search for the nearest instance $x_i^{\mathrm{A,B}}$ in domain B. Both these two searching processes are based on the product of a modified cosine similarity and Euclidean distance,

$$p_{c_i}^{\mathrm{A}} = \arg\min_{p_j^{\mathrm{A}}} \left[ \left( 1 - \frac{f_\theta(x_i^{\mathrm{A}}) \cdot p_j^{\mathrm{A}}}{|f_\theta(x_i^{\mathrm{A}})||p_j^{\mathrm{A}}|} \right) \cdot d(f_\theta(x_i^{\mathrm{A}}), p_j^{\mathrm{A}}) \right] \tag{15}$$

$$x_i^{\mathrm{A,B}} = \arg\min_{x_j^{\mathrm{B}}} \left[ \left( 1 - \frac{f_\theta(x_i^{\mathrm{A}}) \cdot f_\theta(x_j^{\mathrm{B}})}{|f_\theta(x_i^{\mathrm{A}})||f_\theta(x_j^{\mathrm{B}})|} \right) \cdot d(f_\theta(x_i^{\mathrm{A}}), f_\theta(x_j^{\mathrm{B}})) \right]. \tag{16}$$

After obtaining $x_i^{\mathrm{A,B}}$, SN$^2$M searches for its inner nearest prototype $p_{c_i}^{\mathrm{A,B}'}$ in $\mathcal{P}^{\mathrm{B}'}$ (which has been merged with the translated prototype set $\mathcal{P}^{\mathrm{A}\to\mathrm{B}}$ from domain A) and two cases allows us to measure the reliability of $x_i^{\mathrm{A,B}}$. Before introducing these two cases, the inner nearest prototype $p_{c_i}^{\mathrm{A}}$ of $x_i^{\mathrm{A}}$ needs to be translated to domain B and checked whether should be merged to follow condition Eq. (9), and we denote the translated prototype as $\widetilde{p}_{c_i}^{\mathrm{A}}$. Then the first potential case is $p_{c_i}^{\mathrm{A,B}'}$ is exactly identical to $\widetilde{p}_{c_i}^{\mathrm{A}}$, which means, $x_i^{\mathrm{A,B}}$ is convincing since it shares the same prototype correlations with $x_i^{\mathrm{A}}$ across two domains. Then the pair of $x_i^{\mathrm{A,B}}$ and $x_i^{\mathrm{A}}$ should be viewed as a positive pair in the contrastive loss. Otherwise, if $p_{c_i}^{\mathrm{A,B}'}$ is different from $\widetilde{p}_{c_i}^{\mathrm{A}}$, it may be located at the intersection region of multiple clusters. In this case, the pair of $x_i^{\mathrm{A,B}}$ and $x_i^{\mathrm{A}}$ is not supposed to be treated as a positive pair. However, it does not mean SN$^2$M does nothing for these unreliable cases, instead, SN$^2$M leverages a modified prototype contrastive loss to match the pair of $x_i^{\mathrm{A}}$ and $\widetilde{p}_{c_i}^{\mathrm{A}}$. The modification is to incorporate cross-domain instance-wise negative comparison,

$$\mathcal{L}_{\mathrm{SN}^2\mathrm{M}} = \frac{1}{B} \sum_{i=1}^{B} -\log \frac{\Delta}{\sum_{c=1}^{\widetilde{C}^{\mathrm{B}}} \exp(f_\theta(x_i^{\mathrm{A}}) \cdot p_c^{\mathrm{B}'}/\tau) + \sum_{j=1}^{N^{\mathrm{B}}} \exp(f_\theta(x_i^{\mathrm{A}}) \cdot f_\theta(x_j^{\mathrm{B}})/\tau)} \tag{17}$$

$$\text{where } \Delta = \begin{cases} \exp(f_\theta(x_i^{\mathrm{A}}) \cdot \widetilde{p}_{c_i}^{\mathrm{A}}/\tau), \text{ if } p_{c_i}^{\mathrm{A,B}'} \neq \widetilde{p}_{c_i}^{\mathrm{A}} \\ \exp(f_\theta(x_i^{\mathrm{A}}) \cdot \widetilde{p}_{c_i}^{\mathrm{A}}/\tau) + \exp(f_\theta(x_i^{\mathrm{A}}) \cdot f_\theta(x_i^{\mathrm{A,B}})/\tau), \text{ otherwise.} \end{cases}$$

Finally, the overall optimization objective of CDSM follows

$$\mathcal{L}_{\mathrm{CDSM}} = \mathcal{L}_{\mathrm{DAL}} + \mathcal{L}_{\mathrm{SPR}}^{\mathrm{A}} + \mathcal{L}_{\mathrm{SPR}}^{\mathrm{B}} + \mathcal{L}_{\mathrm{SN}^2\mathrm{M}}^{\mathrm{A}} + \mathcal{L}_{\mathrm{SN}^2\mathrm{M}}^{\mathrm{B}}. \tag{18}$$

## 4 Experiments

The datasets, experimental settings, and comparison baselines are introduced below. More implementation details, experiment results, and source codes are provided in the Supplementary Materials.

**Datasets.** *Office-31* [38] includes three domains with 31 classes: Amazon (A), DSLR (D), Webcam (W). *Office-Home* [39] contains four different domains: Art (A), Clipart (C), Product (P), Real (R). And each domain has 67 data categories. *DomainNet* [40] is the most challenging cross-domain dataset to our best knowledge, which includes six domains: Quickdraw (Qu), Clipart (Cl), Painting (Pa), Infograph (In), Sketch (Sk) and Real (Re). DomainNet is originally class-imbalanced, thus we follow [7] to select 7 data classes that contain more than 200 samples.

**Experiment Settings.** For fair comparison, we apply ResNet-50 [41] pre-trained with ImageNet in MoCov2 [10] as the feature extractor. The domain classifier consists of two fully-connected layers.

The SGD optimizer with a momentum of 0.9 is adopted with an initial learning rate of 0.0002 that is scheduled to zero by a cosine learning strategy. The batch size is 64. The training epochs of IDSE are 100 for Office-31 and Office-Home, and 200 for DomainNet. The epoch number of CDSM is 50 for all three datasets. Following [17], we adopt mean average precision on all retrieved results (mAP@All) to measure the performance. All experiments are run repeatedly 3 times with seeds 2024, 2025, and 2026, and we report the mean performance and standard deviation.

**Comparison Baselines.** Our proposed method is compared with a comprehensive set of state-of-the-art works from Cross-Domain Representation Learning (CDS [14], PCS [13]), Unsupervised Domain Generalization (DARL [31], DN2A [32]), and Unsupervised Cross-Domain Retrieval (UCDIR [7], CoDA [17], DGDIR [8]). We follow their default settings and only conduct compulsory customization.

Table 1: Performance comparison (mAP@All) between ours and other baseline methods on Office-31 and DomainNet in Close-set Unsupervised Cross-Domain Retrieval. We blue and bold **the best performance**, and bold **the second best**, same for all tables.

| Methods | A→D | A→W | D→A | D→W | W→A | W→D | Avg. | Qu→Cl | Cl→Pa | Pa→In | In→Sk | Sk→Re | Avg. |
|---|---|---|---|---|---|---|---|---|---|---|---|---|---|
| CDS | $66.7_{\pm1.1}$ | $62.5_{\pm0.9}$ | $70.9_{\pm1.0}$ | $90.0_{\pm0.4}$ | $64.4_{\pm1.5}$ | $88.4_{\pm2.1}$ | $73.8_{\pm0.5}$ | $19.2_{\pm1.0}$ | $35.1_{\pm1.3}$ | $24.4_{\pm0.5}$ | $25.5_{\pm0.7}$ | $32.3_{\pm1.7}$ | $27.3_{\pm0.8}$ |
| PCS | $72.7_{\pm2.2}$ | $70.7_{\pm0.7}$ | $\mathbf{75.3}_{\pm1.4}$ | $88.5_{\pm3.0}$ | $71.2_{\pm1.5}$ | $89.2_{\pm2.6}$ | $77.9_{\pm1.1}$ | $22.2_{\pm0.9}$ | $36.0_{\pm1.3}$ | $27.7_{\pm0.3}$ | $\mathbf{28.0}_{\pm0.5}$ | $33.0_{\pm0.7}$ | $29.4_{\pm0.1}$ |
| DARL | $65.5_{\pm2.5}$ | $70.2_{\pm1.9}$ | $73.4_{\pm3.0}$ | $86.6_{\pm2.2}$ | $69.0_{\pm3.5}$ | $83.7_{\pm2.5}$ | $74.7_{\pm1.2}$ | $22.1_{\pm1.0}$ | $33.7_{\pm1.1}$ | $25.9_{\pm0.8}$ | $27.2_{\pm0.6}$ | $32.5_{\pm1.0}$ | $28.3_{\pm0.4}$ |
| DN2A | $71.1_{\pm1.0}$ | $\mathbf{72.4}_{\pm2.2}$ | $72.5_{\pm1.4}$ | $85.8_{\pm0.7}$ | $71.2_{\pm1.1}$ | $90.0_{\pm1.5}$ | $77.2_{\pm0.8}$ | $23.3_{\pm0.2}$ | $35.0_{\pm0.6}$ | $26.0_{\pm1.1}$ | $27.7_{\pm2.0}$ | $33.0_{\pm1.6}$ | $29.0_{\pm0.7}$ |
| UCDIR | $\mathbf{73.7}_{\pm1.5}$ | $69.9_{\pm3.1}$ | $74.6_{\pm0.4}$ | $91.4_{\pm1.1}$ | $73.1_{\pm0.9}$ | $90.2_{\pm0.6}$ | $78.8_{\pm1.2}$ | $25.8_{\pm1.1}$ | $\mathbf{36.6}_{\pm1.3}$ | $28.0_{\pm0.5}$ | $27.5_{\pm1.3}$ | $\mathbf{34.1}_{\pm0.7}$ | $30.4_{\pm0.4}$ |
| CoDA | $71.7_{\pm2.0}$ | $71.4_{\pm4.3}$ | $74.9_{\pm2.7}$ | $91.4_{\pm1.2}$ | $73.1_{\pm0.9}$ | $90.2_{\pm1.1}$ | $78.8_{\pm1.5}$ | $26.0_{\pm0.8}$ | $34.9_{\pm1.1}$ | $29.2_{\pm0.4}$ | $27.9_{\pm1.0}$ | $33.8_{\pm1.0}$ | $30.4_{\pm0.5}$ |
| DGDIR | $73.5_{\pm2.1}$ | $71.2_{\pm1.4}$ | $75.1_{\pm3.0}$ | $\mathbf{91.7}_{\pm1.1}$ | $\mathbf{74.0}_{\pm1.7}$ | $\mathbf{90.5}_{\pm0.4}$ | $\mathbf{79.3}_{\pm0.6}$ | $\mathbf{27.7}_{\pm0.4}$ | $35.0_{\pm1.5}$ | $\mathbf{30.0}_{\pm2.2}$ | $27.8_{\pm1.5}$ | $33.6_{\pm2.0}$ | $\mathbf{30.8}_{\pm0.9}$ |
| Ours | $\mathbf{76.2}_{\pm1.4}$ | $\mathbf{77.0}_{\pm2.1}$ | $\mathbf{75.6}_{\pm2.0}$ | $\mathbf{92.5}_{\pm0.7}$ | $\mathbf{78.9}_{\pm3.0}$ | $\mathbf{91.0}_{\pm0.2}$ | $\mathbf{81.9}_{\pm0.5}$ | $\mathbf{31.9}_{\pm0.9}$ | $\mathbf{39.4}_{\pm1.4}$ | $\mathbf{35.0}_{\pm0.7}$ | $\mathbf{29.8}_{\pm0.6}$ | $\mathbf{35.7}_{\pm1.8}$ | $\mathbf{34.4}_{\pm0.5}$ |

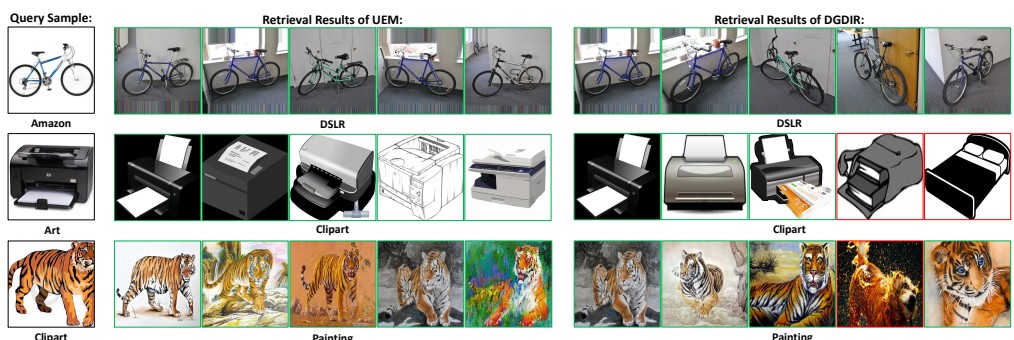

Figure 4: Retrieval results of UEM and DGDIR on Office-31 (A→D), Office-Home (A→C), and DomainNet (Cl→Pa) in Close-set Unsupervised Cross-Domain Retrieval. Green and red rectangles denote correct and incorrect retrieval results. Best viewed in colors.

## 4.1 Effectiveness of UEM When Solving U²CDR

**Close-set Unsupervised Cross-Domain Retrieval.** For the close-set setting, the label space of the query domain is identical to the retrieval domain. All domain pairs of Office-31 and Office-Home are tested, and 5 pairs of DomainNet. The results for Office-31 and DomainNet are shown in Table 1 (Office-Home can be found in the Appendix). According to these results, we can observe that *UEM significantly outperforms other baselines in all cases*. Specifically, we can achieve mAP@All improvement of 2.6% compared to the best baseline method on Office-31, and such improvement is even larger on DomainNet with an average of 3.6%. Figure 4 also shows the retrieval results of our approach on Office-31, Office-Home, and DomainNet, and we can observe that all results are correct.

**Partial Unsupervised Cross-Domain Retrieval.** To establish the partial setting, the query domain contains only half of the label space of the retrieval domain, and the query label space is randomly selected. As shown in Table 2, we can observe that *UEM exceeds all other baseline methods significantly in mAP@All, which is much more substantial than close-set UCDR*. For example, UEM outperforms the second-best with an average of 12.4% on Office-Home. Similar improvements

on Office-31 and DomainNet can be observed (results are provided in the Appendix). Besides, we can easily observe that existing state-of-the-art studies are nearly incapable of dealing with the label space difference in partial UCDR (see Theorem 3.1), while our UEM can work effectively.

**Open-set Unsupervised Cross-Domain Retrieval.** As for the open-set setups, we ensure that the label space of the retrieval domain is half of the query label space. The experiment results for DomainNet (results for Office-31 and Office-Home are in the Appendix) are presented in Table 3 with two metrics – mAP@All for the shared label set, and detection accuracy for the private (open-set) query labels (please refer to Appendix for how to detect the private query labels). According to these results, we can easily observe that ***our approach substantially exceeds other baseline methods in both metrics***. Similar trends can also be found for Office-31 and Office-Home. All these results strongly validate the effectiveness of UEM in open-set UCDR.

Table 2: Performance comparison (mAP@All) between ours and other baseline methods on Office-Home in Partial Unsupervised Cross-Domain Retrieval.

| Methods | A→C | A→P | A→R | C→A | C→P | C→R | P→A | P→C | P→R | R→A | R→C | R→P | Avg. |
|---|---|---|---|---|---|---|---|---|---|---|---|---|---|
| CDS | $22.0_{\pm1.1}$ | $31.1_{\pm0.7}$ | $32.5_{\pm2.0}$ | $26.5_{\pm1.0}$ | $25.6_{\pm0.2}$ | $27.9_{\pm1.5}$ | $30.0_{\pm0.9}$ | $31.8_{\pm1.1}$ | $40.5_{\pm2.7}$ | $32.3_{\pm1.8}$ | $25.5_{\pm1.2}$ | $37.6_{\pm3.0}$ | $30.3_{\pm1.1}$ |
| PCS | $24.5_{\pm0.4}$ | $36.5_{\pm1.2}$ | $38.8_{\pm2.0}$ | $24.9_{\pm1.6}$ | $28.8_{\pm1.1}$ | $29.0_{\pm1.0}$ | $28.6_{\pm2.1}$ | $\mathbf{35.3}_{\pm0.7}$ | $41.7_{\pm1.4}$ | $\mathbf{37.5}_{\pm2.0}$ | $26.9_{\pm1.6}$ | $40.0_{\pm0.9}$ | $32.7_{\pm0.8}$ |
| DARL | $25.5_{\pm1.5}$ | $34.7_{\pm2.0}$ | $29.8_{\pm3.1}$ | $25.0_{\pm1.9}$ | $23.9_{\pm1.7}$ | $27.5_{\pm1.5}$ | $26.8_{\pm2.6}$ | $31.9_{\pm1.1}$ | $40.0_{\pm2.3}$ | $35.5_{\pm1.4}$ | $27.7_{\pm2.0}$ | $40.0_{\pm1.5}$ | $30.7_{\pm1.6}$ |
| DN2A | $\mathbf{25.9}_{\pm0.9}$ | $\mathbf{37.0}_{\pm1.4}$ | $29.5_{\pm2.0}$ | $25.2_{\pm1.0}$ | $27.0_{\pm0.5}$ | $\mathbf{30.5}_{\pm1.1}$ | $29.0_{\pm1.3}$ | $31.5_{\pm0.7}$ | $40.6_{\pm0.4}$ | $35.7_{\pm1.7}$ | $28.0_{\pm0.6}$ | $41.0_{\pm1.1}$ | $31.7_{\pm0.5}$ |
| UCDIR | $23.0_{\pm1.0}$ | $28.7_{\pm2.2}$ | $31.0_{\pm0.9}$ | $26.0_{\pm1.6}$ | $22.0_{\pm1.1}$ | $23.5_{\pm1.6}$ | $\mathbf{31.1}_{\pm1.5}$ | $30.4_{\pm0.2}$ | $40.2_{\pm0.6}$ | $36.9_{\pm1.2}$ | $27.0_{\pm2.1}$ | $36.8_{\pm0.7}$ | $29.7_{\pm0.7}$ |
| CoDA | $22.5_{\pm1.2}$ | $34.2_{\pm1.0}$ | $35.7_{\pm2.0}$ | $25.0_{\pm1.7}$ | $29.5_{\pm0.8}$ | $30.0_{\pm0.9}$ | $30.7_{\pm1.1}$ | $32.0_{\pm1.5}$ | $43.2_{\pm1.3}$ | $35.2_{\pm2.2}$ | $28.5_{\pm1.4}$ | $41.3_{\pm0.3}$ | $32.3_{\pm0.7}$ |
| DGDIR | $24.4_{\pm0.5}$ | $30.9_{\pm2.0}$ | $\mathbf{41.0}_{\pm0.7}$ | $\mathbf{27.2}_{\pm1.2}$ | $\mathbf{30.5}_{\pm2.4}$ | $29.6_{\pm1.7}$ | $30.4_{\pm2.6}$ | $33.2_{\pm1.0}$ | $\mathbf{45.5}_{\pm0.2}$ | $37.1_{\pm1.1}$ | $\mathbf{30.9}_{\pm1.5}$ | $\mathbf{42.0}_{\pm1.0}$ | $\mathbf{33.6}_{\pm0.5}$ |
| Ours | $\mathbf{40.5}_{\pm1.4}$ | $\mathbf{45.8}_{\pm2.0}$ | $\mathbf{48.0}_{\pm2.1}$ | $\mathbf{35.1}_{\pm1.0}$ | $\mathbf{39.2}_{\pm0.5}$ | $\mathbf{41.1}_{\pm0.9}$ | $\mathbf{52.4}_{\pm3.0}$ | $\mathbf{46.0}_{\pm2.1}$ | $\mathbf{55.0}_{\pm1.7}$ | $\mathbf{49.0}_{\pm2.1}$ | $\mathbf{43.1}_{\pm1.0}$ | $\mathbf{56.7}_{\pm1.2}$ | $\mathbf{46.0}_{\pm1.1}$ |

Table 3: Performance comparison (mAP@All for shared-label set, detection accuracy for open-label set) between ours and other baseline methods on DomainNet in Open-set Unsupervised Cross-Domain Retrieval.

| Methods | Qu→Cl | Cl→Pa | Pa→In | In→Sk | Sk→Re | Avg. |
|---|---|---|---|---|---|---|
| | Shared-set mAP@All / Open-set Acc | | | | | |
| CDS | 22.4 58.9 | 34.5 65.2 | 25.5 60.7 | 25.0 59.2 | 33.7 64.9 | 28.2 61.8 |
| PCS | 23.3 57.8 | 34.2 67.8 | 24.9 60.5 | **27.8 65.4** | 34.7 66.9 | 29.0 63.7 |
| DARL | 21.9 54.4 | 32.5 60.2 | 22.0 53.9 | 26.6 60.6 | 32.3 62.8 | 27.1 58.4 |
| DN2A | 22.7 56.6 | 33.4 60.7 | 21.9 55.2 | 24.8 57.8 | 34.0 61.2 | 27.4 58.3 |
| UCDIR | **24.4 59.0** | 34.4 67.0 | 26.7 **62.9** | 25.6 63.4 | **35.5 68.2** | **29.3 64.1** |
| CoDA | 24.2 58.8 | 35.6 66.6 | **27.0** 61.1 | 24.9 58.0 | 34.6 62.9 | **29.3** 61.5 |
| DGDIR | 23.5 57.5 | **36.0 68.3** | 25.8 60.6 | 25.8 59.1 | 35.0 64.3 | 29.2 62.0 |
| Ours | **30.3 72.9** | **40.2 88.1** | **36.0 82.5** | **31.1 78.2** | **36.6 83.0** | **34.8 80.9** |

Table 4: Ablation studies of UEM on Office-31, Office-Home, and DomainNet in Open-set Unsupervised Cross-Domain Retrieval. The average values of Shared-set mAP@All and Open-set Acc for all domain pairs are reported here.

| Variations | Office-31 | | Office-Home | DomainNet | |
|---|---|---|---|---|---|
| | Shared-set mAP@All / Open-set Acc | | | | |
| Ours w/o P.M. | 68.3 | 78.8 | 45.7 72.6 | 30.9 | 70.2 |
| Ours w/o SEL | 72.2 | 88.3 | 47.5 81.7 | 33.0 | 76.4 |
| Ours w/o SPDA | 62.2 | 61.9 | 39.0 60.8 | 24.4 | 60.5 |
| Ours w/o SN$^2$M | 75.0 | 89.2 | 48.8 82.8 | 33.3 | 80.5 |
| Ours | 77.4 | 92.5 | 50.2 86.7 | 34.8 | 80.9 |

## 4.2 Ablation Study

All the ablation studies are carried out in open-set UCDR on three datasets, and the average metrics (Shared-set mAP@All and Open-set Acc) for all domain pairs of a single dataset are reported here.

**Effectiveness of Prototype Merging.** When evaluating the effectiveness of a unified prototypical structure, we do not use prototype merging in IDSE. According to the results of 'Ours w/o P.M.' in Table 4, there is a non-negligible performance drop in both shared-set mAP@All and open-set accuracy. This validates the importance of building a unified prototypical structure across domains.

**Effectiveness of SEL.** When evaluating SEL, we detach it during the model training. According to the results of 'Ours w/o SEL' in Table 4, there is also an evident performance drop compared to the full UEM. This validates that SEL is vital as it can help prepare a better base model for CDSM.

**Effectiveness of SPDA.** We replace our SPDA with the standard domain adversarial learning for the ablation study. As shown in Table 4, there is a significant performance difference between 'Ours w/o SPDA' and the full UEM, which illustrates the importance of semantic preservation during domain alignment, as well as indirectly verifying the necessity of SPDA.

**Effectiveness of** $SN^2M$**.** We also replace $SN^2M$ with the nearest neighboring search approach leveraged by UCDIR. By comparing the results of 'Ours w/o $SN^2M$' and 'Ours', we can conclude that $SN^2M$ is more compatible with UEM and able to achieve more accurate cross-domain categorical matching.

## 5 Conclusion

In this work, we focus on two major challenges when conducting cross-domain retrieval (CDR) in real-world scenarios: one is that the category space across domains is usually distinct, and the other is that both the query and retrieval domains are unlabeled. To tackle these challenges, we propose a Unified, Enhanced, and Matched (UEM) semantic feature learning framework that can establish a unified semantic structure across domains and preserve this structure during categorical matching. Extensive experiments in cases including close-set, partial, open-set unsupervised CDR on multiple datasets demonstrate the effectiveness and universality of UEM, which are reflected in the substantial performance improvement over state-of-the-art studies from Cross-Domain Representation Learning, Unsupervised Domain Generalization, and Unsupervised CDR.

## 6 Limitations and Future Work

To solve the real-world challenges when employing cross-domain retrieval, especially considering the category distinctness across unsupervised data domains, we propose the UEM semantic feature learning framework in this work. Although extensive empirical evaluation and theoretical analysis have validated the effectiveness of UEM, some minor limitations still need more exploration. First, the current UEM framework is composed of two stages, and we empirically determine the switching point. In the future, we need to achieve a real end-to-end UEM by changing the training stage automatically. Besides, there is a lack of theoretical analysis for the second stage (CDSM). In future efforts, we should theoretically prove the semantic preservation of SPDA and the reliability of $SN^2M$. Lastly, the general applicability of UEM also needs testing. For instance, cross-person generalization in wearable devices [42] and property analysis of material [43] or molecular [44] structures require cross-domain retrieval. Therefore, we should test UEM on other modalities like time series and graph data.

## Acknowledgments

We gratefully acknowledge the support of a grant from General Motors.

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

# Appendix

This Appendix includes additional details for the paper *"Semantic Feature Learning for Universal Unsupervised Cross-Domain Retrieval"*, including theoretical proofs (Section A), implementation details (Section B), additional experiment results (Section C), and broader impact (Section D).

## A  Theoretical Proofs

***Theorem** 3.1 (Geometry Distinctness). Suppose data distributions of two domains (A and B) have mutually disjoint supports, and they are uniform over these supports. Assuming the support sets of domains A and B are not identical, the optimal feature extractors $f^*$ that minimize the instance discrimination loss of different domains present distinct geometric feature spaces.*

**Proof.** Suppose a data distribution is made of mutually disjoint supports and distributed uniformly over these supports, there is a theorem demonstrating the representation geometry learned by instance discrimination in a research work [45], i.e.,

**Theorem A.1.** *Assuming a data distribution has mutually disjoint supports and is uniform over these supports, any Simplex ETF representation extracted by $f$ minimizes $\mathcal{L}_{\mathrm{NCE}}(f)$ for any convex and non-increasing loss function $l$. Moreover, if $l$ is strictly convex (e.g., logistic loss), then Simplex ETF representations are the only minimizes of $\mathcal{L}_{\mathrm{NCE}}(f)$.*

The Simplex ETF representations are defined as follows,

**Definition A.2** (Simplex ETF). *A simplex ETF is a collection of equal-length and maximally equiangular vectors. We call a $P \times K$ matrix $\mathbf{M}$ an ETF if it satisfies*

$$\mathbf{M}^{\mathrm{T}}\mathbf{M} = \alpha \left( \frac{K}{K-1}\mathbf{I} - \frac{1}{K-1}\mathbf{1}_K\mathbf{1}_K^{\mathrm{T}} \right), \tag{19}$$

*where $\alpha$ is a non-zero scalar, $\mathbf{I}$ is the identity matrix and $\mathbf{1}_K$ is an all-ones vector.*

In representation learning, ETF representations mean that samples from different categories $(\boldsymbol{x}_i, y_i), (\boldsymbol{x}_j, y_j)$ satisfy

$$\forall y_i \neq y_j, \frac{f_\theta(\boldsymbol{x}_i) \cdot f_\theta(\boldsymbol{x}_j)}{|f_\theta(\boldsymbol{x}_i)||f_\theta(\boldsymbol{x}_j)|} = \frac{-1}{C-1}. \tag{20}$$

With the above preparations, we apply proof by contradiction to prove that different domains hold distinct geometric feature spaces when their respective instance discrimination loss achieves minimization. Specifically, we assume domains A and B share $C$ categories and there is only one category exclusively owned by domain B, i.e.,

$$\mathcal{Y}^{\mathrm{A}} \cap \mathcal{Y}^{\mathrm{B}} = \{y^i\}_{i=1}^C, \ \mathcal{Y}^{\mathrm{B}} \setminus \mathcal{Y}^{\mathrm{A}} = y^{C+1,\mathrm{B}}, \ C^{\mathrm{B}} = C^{\mathrm{A}} + 1. \tag{21}$$

This assumption satisfies that the support sets of domains A and B are not identical. Then suppose that the geometric structures of domains A and B are identical when the instance discrimination loss has been minimized, according to Theorem A.1, both domains A and B present Simplex ETF representations. In this case, if we randomly select two categories $y^p, y^q \in \mathcal{Y}^{\mathrm{A}} \cap \mathcal{Y}^{\mathrm{B}}$ from the domain-shared category set, and their samples within each domain have the following relation,

$$\forall y^p, y^q \in \mathcal{Y}^{\mathrm{A}} \cap \mathcal{Y}^{\mathrm{B}}, \frac{f_\theta(\boldsymbol{x}^{p,\mathrm{A}}) \cdot f_\theta(\boldsymbol{x}^{q,\mathrm{A}})}{|f_\theta(\boldsymbol{x}^{p,\mathrm{A}})||f_\theta(\boldsymbol{x}^{q,\mathrm{A}})|} = \frac{-1}{C^{\mathrm{A}}-1}, \frac{f_\theta(\boldsymbol{x}^{p,\mathrm{B}}) \cdot f_\theta(\boldsymbol{x}^{q,\mathrm{B}})}{|f_\theta(\boldsymbol{x}^{p,\mathrm{B}})||f_\theta(\boldsymbol{x}^{q,\mathrm{B}})|} = \frac{-1}{C^{\mathrm{B}}-1}. \tag{22}$$

According to Eq. (21), Eq. (22) implies that the same category pair across domains has different cosine similarities, which contradicts the assumption of identical geometry across domains.  □

***Theorem** 3.2 (Convergence of IPM). Suppose the data distribution of a domain has mutually disjoint supports, and it is uniform over these supports. Simplex Equiangular Tight Frame (ETF) representations minimize the Instance-Prototype-Mixed Loss of this domain.*

**Proof.** As shown in Eq. (10), the Instance-Prototype-Mixed loss is composed of instance discrimination and prototype contrastive loss. The simplex ETF representations have been proven to minimize instance discrimination in Theorem A.1. Then we only need to prove that the prototype contrastive loss satisfies the convex and non-increasing properties,

**Property A.3.** *For a loss function $l$ defined on a set $\mathcal{V} = \{v_i\}_{i=1}^t$ with the size of $t$, it holds for all subsets of index $\mathcal{S} \subseteq \{1, ..., t\}$ that*

$$l(\mathcal{V}) \geq \frac{1}{|\mathcal{S}|} \sum_{j \in \mathcal{S}} l(\mathcal{V}^{\mathcal{S} \leftarrow j}), \text{ where } \mathcal{V}_i^{\mathcal{S} \leftarrow j} := \begin{cases} v_i, & \text{if } i \notin \mathcal{S} \\ v_j, & \text{otherwise.} \end{cases} \tag{23}$$

To prove the non-increasing property of prototype contrastive loss, we can view $\mathcal{L}_{\text{PNCE}}$ is built on the set $\mathcal{V} = \{v_c = f_\theta(\boldsymbol{x}_i) \cdot \boldsymbol{p}_{c_i}/\tau - f_\theta(\boldsymbol{x}_i) \cdot \boldsymbol{p}_c/\tau\}_{c=1}^C$, then $\mathcal{L}_{\text{PNCE}}(\boldsymbol{x}_i) = l_{\log}(\mathcal{V}) := \log(1 + \sum_c \exp(-v_c))$. We can leverage the concavity of the log function (Jensen's inequality) and denote $T := 1 + \sum_{j \notin \mathcal{S}} \exp(-v_j)$, and we have

$$l_{\log}(\mathcal{V}) = \log(T + \sum_{c \in \mathcal{S}} \exp(-v_c)) \geq \frac{1}{|\mathcal{S}|} \sum_{c \in \mathcal{S}} \log(T + |\mathcal{S}| \exp(-v_c)) = \frac{1}{|\mathcal{S}|} \sum_{c \in \mathcal{S}} l_{\log}(\mathcal{V}^{\mathcal{S} \leftarrow c}). \tag{24}$$

Therefore, the prototype contrastive loss also holds the non-increasing property, which proves that simplex ETF representations minimize the Instance-Prototype-Mixed Loss. □

# B Implementation Details

## B.1 Open-set Query Label Detection

In open-set unsupervised cross-domain retrieval (UCDR) settings, the query domain has some private categories that are not included in the retrieval domain. In this case, the retrieval results of query samples for such private query categories should be null. As aforementioned, for a query sample $\boldsymbol{x}_i^{\text{A}}$, the retrieval process needs to calculate the distance between all samples in the retrieval domain and $\boldsymbol{x}_i^{\text{A}}$. Then the most similar retrieval samples are supposed to be those located as close to $\boldsymbol{x}_i^{\text{A}}$ as possible. Intuitively, if $\boldsymbol{x}_i^{\text{A}}$ belongs to the private query categories, the nearest sample in the retrieval domain should be relatively distant. Therefore, there is a need for a threshold that allows us to determine whether the closest retrieval sample of a query sample is located too far to be a similar sample. Next, let us introduce how our UEM detects and identifies whether a query sample belongs to private query categories.

In our UEM framework, the crucial design is to build a unified prototypical structure across domains, which also shapes the private label detection strategy. Specifically, after the training of CDSM, we apply K-Means to the query and retrieval domain datasets again to construct the prototype sets $\mathcal{P}^{\text{A}}$ and $\mathcal{P}^{\text{B}}$. Then the prototypes of the retrieval domain are translated to the query domain for potential merging. The detailed prototype translation and merging have been introduced in Section 3.2.1. After the prototype merging, both $\mathcal{P}^{\text{A}}$ and $\mathcal{P}^{\text{B}}$ are divided into two groups by satisfying the merging condition (Eq. (9)) or not. For the prototype pairs that satisfy the merging condition, we record the maximum inter-sample distance among their clusters as

$$D_{c\oplus} = \max_{\boldsymbol{x}_i^{\text{A}} \in \mathcal{X}_{c\oplus}^{\text{A}}, \boldsymbol{x}_j^{\text{B}} \in \mathcal{X}_{c\oplus}^{\text{B}}} \left[ \left( 1 - \frac{f_\theta(\boldsymbol{x}_i^{\text{A}}) \cdot f_\theta(\boldsymbol{x}_j^{\text{B}})}{|f_\theta(\boldsymbol{x}_i^{\text{A}})||f_\theta(\boldsymbol{x}_j^{\text{B}})|} \right) \cdot d(f_\theta(\boldsymbol{x}_i^{\text{A}}), f_\theta(\boldsymbol{x}_j^{\text{B}})) \right] \tag{25}$$

$$\mathcal{X}_{c\oplus}^{\text{A}} = \left\{ \boldsymbol{x}_i^{\text{A}} \middle| \arg\min_{\boldsymbol{p}^{\text{A}}} \left[ d(f_\theta(\boldsymbol{x}_i^{\text{A}}), \boldsymbol{p}^{\text{A}}) \right] = \boldsymbol{p}_{c\oplus}^{\text{A},\oplus} \right\}, \mathcal{X}_{c\oplus}^{\text{B}} = \left\{ \boldsymbol{x}_j^{\text{B}} \middle| \arg\min_{\boldsymbol{p}^{\text{B}}} \left[ d(f_\theta(\boldsymbol{x}_j^{\text{B}}), \boldsymbol{p}^{\text{B}}) \right] = \boldsymbol{p}_{c\oplus}^{\text{B},\oplus} \right\} \tag{26}$$

Then for any query sample $\boldsymbol{x}_i^{\text{A}}$, there are two cases for its belonging. One is $\boldsymbol{x}_i^{\text{A}}$ belongs to the clusters of prototypes unmerged, i.e., $\mathcal{P}^{\text{A}} \setminus \mathcal{P}^{\text{A},\oplus}$. In this case, $\boldsymbol{x}_i^{\text{A}}$ is supposed to come from private query labels with high confidence. The other case is that the closest prototype of $\boldsymbol{x}_i^{\text{A}}$ satisfies the merging condition, i.e., $\arg\min_{\boldsymbol{p}^{\text{A}}} \left[ d(f_\theta(\boldsymbol{x}_i^{\text{A}}), \boldsymbol{p}^{\text{A}}) \right] = \boldsymbol{p}_{c\oplus}^{\text{A},\oplus} \in \mathcal{P}^{\text{A},\oplus}$, in which we should compare the recorded $D_{c\oplus}$ and the minimal distance between all samples in the retrieval domain and $\boldsymbol{x}_i^{\text{A}}$, i.e.,

$$D_i^{\text{A} \to \text{B}} = \min_{\boldsymbol{x}_j^{\text{B}} \in \mathcal{D}^{\text{B}}} \left[ \left( 1 - \frac{f_\theta(\boldsymbol{x}_i^{\text{A}}) \cdot f_\theta(\boldsymbol{x}_j^{\text{B}})}{|f_\theta(\boldsymbol{x}_i^{\text{A}})||f_\theta(\boldsymbol{x}_j^{\text{B}})|} \right) \cdot d(f_\theta(\boldsymbol{x}_i^{\text{A}}), f_\theta(\boldsymbol{x}_j^{\text{B}})) \right]. \tag{27}$$

If $D_{c\oplus} < D_i^{A\rightarrow B}$, we have faith that there is no sample similar enough in the retrieval domain, which means $x_i^A$ should belong to private query categories. Otherwise, $x_i^A$ comes from the shared label set, and we should conduct the normal retrieval operation.

## B.2 Comparison Baseline Implementation

In our evaluation process, we implement a number of state-of-the-art baseline methods to compare our proposed UEM in U$^2$CDR. For a fair comparison, we ensure two principles for all these used baselines – one is that the training data consists of at least two domains, and the other is that the training data is unlabeled. Following these two principles, in addition to unsupervised cross-domain retrieval studies [7, 8, 17], two other problems share similar setups: cross-domain representation learning (CDRL) [14, 13] and unsupervised domain generalization (UDG) [31, 32]. For CDRL, the objective is to learn domain-generalizable representations that provide domain-transferable knowledge for any downstream task. One of the most typical downstream tasks of CDRL is cross-domain retrieval. As for UDG, in addition to achieving effective cross-domain representation learning, domain-generalizable classifiers are also needed. However, in our evaluation, the retrieval process does not require any classifier, thus we omit all techniques related to classifier training for the used UDG approaches.

For the specific setups of our experiments, we consider close-set, partial, and open-set UCDR. The close-set UCDR assumes that the label spaces of the query and retrieval domains are identical, which is the benchmark setup of other baseline methods. In this case, we follow the default settings of these baseline methods to evaluate their performance in close-set UCDR. As for the partial UCDR, the label space of the query domain is half of the retrieval label space, where most baseline methods can work normally without any modification. But some baseline approaches (PCS [13], UCDIR [7], DGDIR [8]), especially those based on prototype learning, require the knowledge of category numbers for domains, therefore, we suppose the category numbers are known to these approaches. The last open-set UCDR supposes the retrieval label space is half of the query label space. In this setup, the query domain has private categories, and if we conduct retrieval for samples from these categories, the retrieval results should be null. To detect private categories, we follow the strategy leveraged by UEM (Section B.1) to employ a similar one for the used baseline methods. Specifically, we divide all baseline methods into two groups by whether there is a dedicated nearest neighboring searching algorithm. For those (DARL [31]) that don't have the nearest neighboring search, we attach the searching algorithm used by UCDIR [7]. After conducting all training and operations of any baseline method, we use the nearest neighboring search to pair samples from the query and retrieval domains. Moreover, we conduct K-Means in the query domain to build the prototype set $\mathcal{P}^A$ (for CoDA [17], we leverage its auxiliary classifiers to construct the prototype set). Then for each cluster in the query domain, we record the maximum inter-sample distance $D_c^A$. Note that the distance measurement here is different from the product of minus cosine similarity and Euclidean distance (Eq. (25)), and different approaches use diverse measurement, e.g., UCDIR [7] uses cosine similarity while DN2A [32] leverages Euclidean distance.

Table 5: Performance comparison (mAP@All) between ours and other baseline methods on Office-Home in Close-set Unsupervised Cross-Domain Retrieval.

| Methods | A→C | A→P | A→R | C→A | C→P | C→R | P→A | P→C | P→R | R→A | R→C | R→P | Avg. |
|---|---|---|---|---|---|---|---|---|---|---|---|---|---|
| CDS | $33.0_{\pm0.3}$ | $44.5_{\pm1.1}$ | $51.4_{\pm2.3}$ | $32.4_{\pm0.9}$ | $40.3_{\pm1.9}$ | $41.8_{\pm2.0}$ | $45.3_{\pm1.5}$ | $41.5_{\pm1.6}$ | $60.8_{\pm0.8}$ | $51.1_{\pm2.9}$ | $42.0_{\pm1.8}$ | $58.8_{\pm1.0}$ | $45.2_{\pm1.2}$ |
| PCS | $34.3_{\pm1.1}$ | $46.3_{\pm1.4}$ | $51.6_{\pm0.5}$ | $32.3_{\pm2.0}$ | $40.5_{\pm1.1}$ | $40.6_{\pm0.6}$ | $47.0_{\pm0.6}$ | $42.1_{\pm1.5}$ | $61.3_{\pm2.5}$ | $51.6_{\pm2.7}$ | $42.8_{\pm1.9}$ | $60.1_{\pm1.3}$ | $45.9_{\pm0.9}$ |
| DARL | $32.4_{\pm1.0}$ | $40.9_{\pm2.2}$ | $50.5_{\pm2.0}$ | $33.0_{\pm1.9}$ | $36.7_{\pm0.7}$ | $41.5_{\pm1.7}$ | $47.0_{\pm2.5}$ | $40.9_{\pm1.4}$ | $59.0_{\pm0.9}$ | $51.1_{\pm1.2}$ | $43.0_{\pm2.0}$ | $60.8_{\pm3.1}$ | $44.7_{\pm1.4}$ |
| DN2A | $35.5_{\pm0.5}$ | $42.8_{\pm1.7}$ | $52.9_{\pm2.6}$ | $34.0_{\pm1.4}$ | $35.7_{\pm1.1}$ | $42.0_{\pm1.0}$ | $48.0_{\pm2.7}$ | $43.2_{\pm1.6}$ | $59.8_{\pm1.4}$ | $49.0_{\pm2.3}$ | $44.7_{\pm1.1}$ | $56.5_{\pm2.0}$ | $45.3_{\pm1.3}$ |
| UCDIR | $\mathbf{36.1}_{\pm1.5}$ | $46.5_{\pm0.9}$ | $\mathbf{55.9}_{\pm1.2}$ | $34.0_{\pm1.8}$ | $44.1_{\pm1.3}$ | $43.1_{\pm2.0}$ | $51.2_{\pm1.1}$ | $44.1_{\pm1.4}$ | $\mathbf{67.1}_{\pm2.3}$ | $52.7_{\pm1.9}$ | $43.0_{\pm3.7}$ | $66.5_{\pm0.4}$ | $48.7_{\pm1.6}$ |
| CoDA | $34.7_{\pm0.8}$ | $49.6_{\pm1.0}$ | $53.2_{\pm0.9}$ | $33.2_{\pm1.1}$ | $42.9_{\pm2.6}$ | $\mathbf{44.7}_{\pm1.5}$ | $50.4_{\pm2.5}$ | $45.2_{\pm2.2}$ | $65.2_{\pm1.0}$ | $\mathbf{53.1}_{\pm0.9}$ | $\mathbf{46.0}_{\pm2.4}$ | $65.2_{\pm1.6}$ | $46.8_{\pm1.2}$ |
| DGDIR | $36.0_{\pm1.0}$ | $\mathbf{50.1}_{\pm1.7}$ | $55.5_{\pm0.4}$ | $\mathbf{34.1}_{\pm1.1}$ | $\mathbf{44.9}_{\pm0.9}$ | $43.0_{\pm2.2}$ | $\mathbf{51.5}_{\pm1.7}$ | $\mathbf{45.5}_{\pm1.9}$ | $66.6_{\pm2.2}$ | $53.0_{\pm2.0}$ | $44.5_{\pm1.1}$ | $\mathbf{67.0}_{\pm1.3}$ | $\mathbf{49.3}_{\pm1.4}$ |
| Ours | $\mathbf{38.5}_{\pm1.5}$ | $\mathbf{52.6}_{\pm0.9}$ | $\mathbf{59.0}_{\pm1.2}$ | $\mathbf{34.2}_{\pm0.3}$ | $\mathbf{47.5}_{\pm2.0}$ | $\mathbf{49.0}_{\pm1.7}$ | $\mathbf{55.2}_{\pm1.3}$ | $\mathbf{49.0}_{\pm1.1}$ | $\mathbf{69.5}_{\pm1.7}$ | $\mathbf{56.9}_{\pm2.1}$ | $\mathbf{48.7}_{\pm0.4}$ | $\mathbf{68.2}_{\pm1.0}$ | $\mathbf{52.4}_{\pm0.7}$ |

## C   Additional Experiments

Here we provide the additional experiment results including the close-set UCDR on Office-Home (Table 5), partial UCDR on Office-31 and DomainNet (Table 6), and open-set UCDR on Office-31 (Table 7) and Office-Home (Table 8). According to these results, we can obtain similar observations and conclusions to the main paper. First, our UEM can achieve the best performance in close-set

Table 6: Performance comparison (mAP@All) between ours and other baseline methods on Office-31 and DomainNet in Partial Unsupervised Cross-Domain Retrieval.

| Methods | A→D | A→W | D→A | D→W | W→A | W→D | Avg. | Qu→Cl | Cl→Pa | Pa→In | In→Sk | Sk→Re | Avg. |
|---|---|---|---|---|---|---|---|---|---|---|---|---|---|
| CDS | $42.5_{\pm1.2}$ | $39.7_{\pm1.4}$ | $45.4_{\pm2.0}$ | $\mathbf{68.8}_{\pm0.6}$ | $40.7_{\pm0.7}$ | $55.8_{\pm1.1}$ | $48.8_{\pm0.5}$ | $17.7_{\pm1.9}$ | $31.1_{\pm1.2}$ | $22.0_{\pm2.3}$ | $21.8_{\pm1.1}$ | $29.6_{\pm0.6}$ | $24.4_{\pm1.0}$ |
| PCS | $44.0_{\pm2.3}$ | $41.5_{\pm1.4}$ | $47.9_{\pm1.9}$ | $65.9_{\pm2.2}$ | $47.3_{\pm0.9}$ | $56.5_{\pm0.2}$ | $50.5_{\pm1.2}$ | $18.0_{\pm0.8}$ | $\mathbf{31.5}_{\pm1.3}$ | $23.4_{\pm1.8}$ | $22.1_{\pm0.7}$ | $27.9_{\pm0.5}$ | $24.6_{\pm0.4}$ |
| DARL | $41.5_{\pm1.0}$ | $38.9_{\pm2.2}$ | $49.0_{\pm2.5}$ | $65.5_{\pm3.3}$ | $45.7_{\pm1.9}$ | $53.6_{\pm2.4}$ | $49.0_{\pm1.8}$ | $19.0_{\pm1.1}$ | $30.5_{\pm2.0}$ | $23.3_{\pm1.6}$ | $23.0_{\pm2.2}$ | $28.7_{\pm1.3}$ | $24.9_{\pm1.2}$ |
| DN2A | $42.5_{\pm1.0}$ | $40.0_{\pm2.6}$ | $\mathbf{51.1}_{\pm0.5}$ | $66.7_{\pm1.5}$ | $46.5_{\pm0.9}$ | $55.0_{\pm1.3}$ | $50.3_{\pm1.1}$ | $18.7_{\pm0.4}$ | $31.1_{\pm1.3}$ | $25.0_{\pm0.7}$ | $24.4_{\pm1.4}$ | $30.7_{\pm1.8}$ | $26.0_{\pm0.9}$ |
| UCDIR | $\mathbf{46.0}_{\pm1.4}$ | $\mathbf{41.8}_{\pm2.0}$ | $50.3_{\pm1.2}$ | $62.9_{\pm2.1}$ | $47.0_{\pm1.4}$ | $56.9_{\pm2.2}$ | $50.8_{\pm1.5}$ | $19.3_{\pm1.3}$ | $29.8_{\pm0.8}$ | $24.4_{\pm1.4}$ | $23.1_{\pm0.7}$ | $29.0_{\pm1.1}$ | $25.1_{\pm0.8}$ |
| CoDA | $45.1_{\pm2.0}$ | $40.7_{\pm1.2}$ | $49.3_{\pm3.0}$ | $66.0_{\pm1.4}$ | $47.0_{\pm1.6}$ | $\mathbf{57.8}_{\pm0.3}$ | $\mathbf{51.0}_{\pm1.3}$ | $\mathbf{20.2}_{\pm0.3}$ | $30.9_{\pm1.0}$ | $\mathbf{25.2}_{\pm1.1}$ | $24.0_{\pm0.9}$ | $31.0_{\pm0.5}$ | $\mathbf{26.3}_{\pm0.6}$ |
| DGDIR | $45.0_{\pm1.2}$ | $39.2_{\pm0.6}$ | $49.7_{\pm1.3}$ | $64.4_{\pm2.0}$ | $\mathbf{48.5}_{\pm1.0}$ | $55.2_{\pm1.3}$ | $50.3_{\pm0.8}$ | $18.8_{\pm0.5}$ | $30.4_{\pm1.6}$ | $25.0_{\pm0.2}$ | $\mathbf{24.5}_{\pm0.8}$ | $\mathbf{31.2}_{\pm1.0}$ | $26.0_{\pm0.5}$ |
| Ours | $\mathbf{64.4}_{\pm1.6}$ | $\mathbf{51.0}_{\pm2.2}$ | $\mathbf{59.4}_{\pm1.4}$ | $\mathbf{76.9}_{\pm2.2}$ | $\mathbf{61.5}_{\pm1.2}$ | $\mathbf{65.0}_{\pm2.0}$ | $\mathbf{63.0}_{\pm1.7}$ | $\mathbf{28.2}_{\pm1.2}$ | $\mathbf{34.9}_{\pm0.4}$ | $\mathbf{32.7}_{\pm0.7}$ | $\mathbf{26.6}_{\pm1.5}$ | $\mathbf{34.0}_{\pm0.9}$ | $\mathbf{31.3}_{\pm0.8}$ |

Table 7: Performance comparison (mAP@All for shared-label set, detection accuracy for open-label set) between ours and other baselines on Office-31 in Open-set Unsupervised Cross-Domain Retrieval.

| Methods | A→D | | A→W | | D→A | | D→W | | W→A | | W→D | | Avg. | |
|---|---|---|---|---|---|---|---|---|---|---|---|---|---|---|
| | Shared-set mAP@All / Open-set Acc | | | | | | | | | | | | | |
| CDS | $60.7_{\pm0.8}$ | $76.2_{\pm2.3}$ | $56.8_{\pm1.3}$ | $80.2_{\pm1.9}$ | $62.9_{\pm0.6}$ | $81.3_{\pm1.7}$ | $80.7_{\pm1.4}$ | $90.2_{\pm2.7}$ | $60.2_{\pm1.5}$ | $79.0_{\pm3.0}$ | $80.3_{\pm1.0}$ | $89.7_{\pm2.2}$ | $66.9_{\pm0.7}$ | $82.8_{\pm2.0}$ |
| PCS | $62.5_{\pm1.4}$ | $78.9_{\pm2.2}$ | $\mathbf{67.3}_{\pm0.6}$ | $\mathbf{84.4}_{\pm1.0}$ | $65.8_{\pm1.2}$ | $\mathbf{82.9}_{\pm3.4}$ | $82.9_{\pm0.9}$ | $\mathbf{90.8}_{\pm1.5}$ | $63.3_{\pm0.9}$ | $79.5_{\pm1.9}$ | $81.6_{\pm1.8}$ | $\mathbf{91.1}_{\pm2.9}$ | $\mathbf{70.6}_{\pm1.1}$ | $\mathbf{84.6}_{\pm1.8}$ |
| DARL | $60.9_{\pm1.2}$ | $78.0_{\pm2.4}$ | $60.5_{\pm0.9}$ | $77.2_{\pm1.9}$ | $62.6_{\pm1.1}$ | $78.0_{\pm1.5}$ | $78.3_{\pm1.4}$ | $88.4_{\pm3.0}$ | $60.0_{\pm1.3}$ | $79.4_{\pm2.7}$ | $78.1_{\pm0.5}$ | $87.9_{\pm1.3}$ | $66.7_{\pm0.8}$ | $81.5_{\pm2.1}$ |
| DN2A | $61.7_{\pm0.4}$ | $76.9_{\pm1.6}$ | $60.8_{\pm1.1}$ | $75.3_{\pm2.3}$ | $62.5_{\pm0.5}$ | $78.0_{\pm1.0}$ | $80.2_{\pm2.3}$ | $90.4_{\pm5.6}$ | $61.1_{\pm1.0}$ | $76.7_{\pm2.6}$ | $78.8_{\pm0.7}$ | $90.0_{\pm1.8}$ | $67.5_{\pm1.2}$ | $81.2_{\pm2.4}$ |
| UCDIR | $62.9_{\pm0.6}$ | $78.1_{\pm1.4}$ | $65.5_{\pm0.9}$ | $82.4_{\pm1.6}$ | $62.6_{\pm2.0}$ | $80.4_{\pm4.7}$ | $79.9_{\pm1.3}$ | $86.9_{\pm2.2}$ | $64.4_{\pm0.9}$ | $84.3_{\pm1.7}$ | $80.5_{\pm0.9}$ | $\mathbf{91.1}_{\pm2.0}$ | $69.3_{\pm0.9}$ | $83.9_{\pm1.6}$ |
| CoDA | $60.9_{\pm1.1}$ | $74.6_{\pm2.0}$ | $62.4_{\pm1.3}$ | $78.0_{\pm1.9}$ | $\mathbf{66.6}_{\pm0.6}$ | $82.2_{\pm1.4}$ | $\mathbf{83.5}_{\pm1.3}$ | $90.0_{\pm3.8}$ | $\mathbf{70.3}_{\pm0.2}$ | $\mathbf{85.0}_{\pm1.1}$ | $78.5_{\pm1.0}$ | $89.0_{\pm2.4}$ | $70.4_{\pm0.9}$ | $83.1_{\pm1.9}$ |
| DGDIR | $\mathbf{65.0}_{\pm0.9}$ | $\mathbf{81.1}_{\pm1.7}$ | $62.3_{\pm1.6}$ | $80.0_{\pm2.9}$ | $64.0_{\pm0.7}$ | $79.9_{\pm1.7}$ | $80.9_{\pm1.1}$ | $87.5_{\pm2.0}$ | $66.8_{\pm0.3}$ | $79.4_{\pm1.2}$ | $\mathbf{82.2}_{\pm1.8}$ | $91.0_{\pm3.5}$ | $70.2_{\pm1.1}$ | $83.2_{\pm2.2}$ |
| Ours | $\mathbf{70.6}_{\pm1.2}$ | $\mathbf{90.7}_{\pm1.9}$ | $\mathbf{76.8}_{\pm0.4}$ | $\mathbf{93.0}_{\pm1.5}$ | $\mathbf{72.4}_{\pm2.0}$ | $\mathbf{89.8}_{\pm3.9}$ | $\mathbf{87.7}_{\pm1.7}$ | $\mathbf{95.5}_{\pm3.2}$ | $\mathbf{74.0}_{\pm1.1}$ | $\mathbf{90.2}_{\pm2.0}$ | $\mathbf{82.9}_{\pm1.9}$ | $\mathbf{95.8}_{\pm3.3}$ | $\mathbf{77.4}_{\pm1.4}$ | $\mathbf{92.5}_{\pm2.5}$ |

UCDR, which is reflected by the average performance improvement of 3.1% on Office-Home. Moreover, UEM can exceed all other baseline methods much more substantially in both partial and open-set UCDR. Specifically, our methods outperform the best baseline with a margin of 5.0% on DomainNet and 12.0% on Office-31 in partial UCDR. As for open-set UCDR, UEM exceeds other baseline methods with a range of 6.8% and 7.9% on Office-31 for shared-set mAP@All and open-set detection accuracy respectively, and such improvement is much higher on Office-Home in 15.1% for shared-set mAP@All and 13.1% for open-set detection accuracy. In addition, if we compare the results of the same domain pairs between close-set and partial/open-set UCDR for baseline methods, we can observe that the performance drops a lot, which also validates the existence of geometry distinctness (Theorem 3.1). In particular, such geometry distinctness originating from the category space difference can incur a performance drop of up to 29% for DGDIR on Office-31 between close-set and partial UCDR.

# D  Broader Impact

The research development for solving Universal Unsupervised Cross-Domain Retrieval ($\text{U}^2\text{CDR}$) has the potential to make a significant positive impact across various sectors. By enabling more accurate and flexible retrieval of information across different domains without the need for supervision and the concern about semantic category distinctness, our proposed UEM framework can enhance user experiences in product recommendation systems, leading to more personalized and relevant suggestions. In the realm of artistic creation, UEM can facilitate novel connections and inspirations by retrieving cross-domain artistic elements, fostering creativity and innovation. Beyond these applications, UEM can also be beneficial in healthcare, finance, and scientific research with certain adaptive modifications. In healthcare, it can improve diagnostic tools and personalized treatment plans by integrating diverse data sources. In finance, it can enhance risk assessment and fraud detection by analyzing cross-domain financial data. In scientific research, it can accelerate discoveries by connecting insights from different fields. While UEM has substantial benefits, it is crucial that its deployment adheres to existing privacy and intellectual property protection regulations and policies. Ensuring that data used in cross-domain retrieval respects user privacy and intellectual property rights is essential to prevent misuse and maintain public trust. Thus, applying this research in the real world should consider ethical issues to ensure responsible and fair use, thereby aiming for a positive societal impact without any negative social consequences.

Table 8: Performance comparison (mAP@All for shared-label set, detection accuracy for open-label set) between ours and others on Office-Home in Open-set Unsupervised Cross-Domain Retrieval.

| Methods | A→C | A→P | A→R | C→A | C→P | C→R | P→A | P→C | P→R | R→A | R→C | R→P | Avg. |
|---|---|---|---|---|---|---|---|---|---|---|---|---|---|
| | Shared-set mAP@All / Open-set Acc | | | | | | | | | | | | |
| CDS | 26.9 60.2 | 34.7 68.7 | 33.9 67.5 | 27.9 66.2 | 28.8 65.5 | 30.0 66.2 | **32.1** 68.9 | 32.5 70.4 | 44.2 74.5 | 32.5 69.0 | 26.7 65.4 | 40.0 76.0 | 32.5 68.2 |
| PCS | 26.6 61.1 | 37.8 72.0 | 40.9 76.6 | **29.0** 65.4 | 30.1 69.9 | **34.2** 71.9 | 31.5 70.8 | **37.9** 74.0 | 42.6 77.4 | **38.9** 72.8 | 29.4 66.6 | 42.7 78.0 | **35.1** 71.4 |
| DARL | 26.9 **64.0** | 36.9 70.1 | 28.5 66.4 | 26.7 65.0 | 22.0 60.6 | 29.4 68.2 | 25.9 64.3 | 33.0 69.1 | 42.5 79.5 | 35.0 72.2 | 28.0 65.1 | 42.2 78.0 | 31.4 68.5 |
| DN2A | **27.0** 63.3 | **38.2 74.4** | 29.0 68.8 | 26.8 65.4 | 28.9 67.5 | 34.0 **72.4** | 29.0 66.8 | 33.0 71.5 | 42.0 80.3 | 35.5 72.6 | 29.4 68.8 | 41.5 79.4 | 32.9 70.9 |
| UCDIR | 24.7 62.2 | 30.1 64.4 | 26.8 64.0 | 26.5 65.2 | 21.5 62.0 | 25.5 67.0 | 32.0 **71.9** | 31.1 70.8 | 43.0 78.9 | 37.6 76.6 | 28.0 69.2 | 38.5 77.4 | 30.4 69.1 |
| CoDA | 24.9 63.0 | 35.5 74.3 | 36.0 74.5 | 25.5 65.9 | 31.1 **70.8** | 32.5 72.3 | 30.5 71.1 | 33.7 **74.4** | 44.4 **81.5** | 37.0 **78.3** | **36.6 76.0** | 42.1 80.9 | 34.2 **73.6** |
| DGDIR | 25.5 62.5 | 31.7 67.7 | **41.5 78.9** | 27.8 **67.5** | **31.2** 68.1 | 29.0 66.5 | 31.0 68.3 | 33.0 68.9 | **45.8** 80.9 | 37.0 76.6 | 31.5 68.0 | **44.0 82.2** | 34.1 71.3 |
| Ours | **40.6 80.8** | **49.0 87.7** | **55.4 92.0** | **33.7 72.2** | **45.0 85.3** | **47.7 87.9** | **53.5 90.2** | **48.7 87.8** | **64.4 90.0** | **52.2 90.3** | **47.7 86.7** | **64.5 89.5** | **50.2 86.7** |

