# OpenReview forum: "Semantic Feature Learning for Universal Unsupervised Cross-Domain Retrieval"
_NeurIPS.cc/2024/Conference — NeurIPS 2024 poster_

### Official Review · Reviewer_ssYY · 2024-07-11

**Soundness:** 3
**Presentation:** 3
**Contribution:** 3
**Rating:** 7
**Confidence:** 4

**Summary:**

This paper introduces the problem of Universal Unsupervised Cross-Domain Retrieval (U2CDR) and proposes a two-stage semantic feature learning framework to address it. The framework includes a cross-domain unified prototypical structure established through an instance-prototype-mixed contrastive loss and a semantic-enhanced loss in the first stage, and a modified adversarial training mechanism to ensure minimal changes during domain alignment in the second stage. Extensive experiments demonstrate that this approach significantly outperforms existing state-of-the-art CDR methods in solving U2CDR challenges.

**Strengths:**

1. This paper addresses a new problem, namely Universal Unsupervised Cross-Domain Retrieval, and proposes an initial solution.
2. The paper first formulates the problem and then introduces the proposed method in a hierarchical manner, which is clear and well-structured.
3. The ability to perform U2CDR has broad implications for various applications, such as image search, product recommendations, and artistic creation.

**Weaknesses:**

1. The main effort of the paper seems to be on designing an optimization method. However, the optimization methods involved appear to be mostly existing ones. The authors should enhance the description of the novelty.
2. Although the paper uses $L_{SPR}$ to maintain the semantic structure within domains, how to maintain the relationship between the positive pairs across domains should be emphasized.
3. The analysis related to the Ablation Study seems insufficient. It would be beneficial to analyze the reasons for the experimental results in Table 4.

**Questions:**

While this paper introduces a new problem, where exactly is the novelty in the methodology section?

**Limitations:**

No limitations or negative impacts have been identified in this paper.

---

> ### Author Rebuttal · Authors · 2024-08-07
>
> > Description of the methodology novelty
>
> Please refer to the novelty illustration in the Global Response. In addition to the ablation study, To further validate the novelty of semantic structure preservation and cross-domain matching, we carry out experiments with the replacement of another state-of-the-art binary graph-based semantic preservation approach (SPPA [1]) and nearest neighbor searching algorithm (CD^2NN [2]). SPPA preserves the instance-level cosine similarity within each cluster and the prototype-level Euclidean distance across clusters. CD^2NN determines cross-domain nearest instance pair by seeking neighboring instance consistency. The experiment results in open-set CDR (in each cell, the left number is the Shared-set mAP@All, while the right is the Open-set detection accuracy) are shown below, demonstrating that UEM's semantic structure preservation and cross-domain matching are more effective.
>
> |         | Office-31  | Office-Home | DomainNet  |
> |---------|------------|-------------|------------|
> | UEM w/ SPPA  | 76.0, 88.9| 47.2, 84.4  | 31.5, 75.7   |
> | UEM w/ CD^2NN | 76.4, 89.4 |   46.5, 85.0 | 32.1, 76.4 |
> | UEM     | 77.4, 92.5 |  50.2, 86.7  | 34.8, 80.9 |
>
> [1] Lin, et al. Continual semantic segmentation via structure preserving and projected feature alignment. ECCV 2022
>
> [2] Liu, et al. Promoting Semantic Connectivity: Dual Nearest Neighbors Contrastive Learning for Unsupervised Domain Generalization. CVPR 2023
>
> > How does UEM maintain the relationship between positive pairs across domains
>
> Thank you for your suggestion. In fact, **cross-domain positive pairs are maintained through the construction of that unified prototypical structure.** During the first stage of our UEM framework, all instances in each domain approach their nearest prototype. These prototypes undergo a prototype conversion process, making them cross-domain unified, meaning that prototypes of the same category have the same geometric relationship relative to other categories within each domain. **In this scenario, cross-domain positive pairs will be close to the prototype of the same category in their respective domains.**
>
> The relationship of cross-domain positive pairs is preserved during the second stage of UEM. As the prototypical structures across domains are unified, **the inner-domain relationship preservation achieved by our semantic-preserving domain alignment actually equals to preserving inter-domain relationships, which definitely includes the relationship of cross-domain positive pairs. Besides, the relationship of cross-domain positive pairs is further reinforced in SN^2M,** which is a primary objective of cross-domain matching.
>
> > More analysis of the ablation study
>
> Thanks for your suggestion. We provide more analysis and reasoning for the ablation study as follows, and will incorporate them into the future revision.
>
> The performance degradation caused by not using Prototype Merging may be attributed to the geometry distinctness of instance discrimination learning within each domain (Theorem 3.1). The high-level intuition is that the same categories across domains are forced to distinct geometry locations due to distinct contrastive comparisons with different category spaces. In this case, there are plenty of mismatches and misalignments in cross-domain matching. Therefore, the UEM performs poorly without using Prototype Merging.
>
> As for the performance loss due to SEL, we think it originated from the unavoidable errors of the arbitrary prototype allocation in the prototype contrastive loss. Different from this single prototype allocation, SEL considers the potential relationship between the instance and all prototypes. This design provides the possibility of error correction and can effectively compensate for the prototype contrastive loss.
>
> According to Table 4, there is the largest performance gap between the 'Ours w/o SPDA' approach and the full 'Ours' approach. This indicates the semantic structure destruction during standard domain adversarial training. The semantic structure learned by the first stage is the basis of cross-domain matching, thus causing such a large performance drop.
>
> To test the effectiveness of SN^2M, we replace it with the neighbor searching used in UCDIR, which searches for the nearest cross-domain instance and prototype in terms of cosine similarity distance to approach. Apparently, this search strategy has a lot of errors, which is worse if there is a domain gap. By contrast, our SN^2M can measure the reliability of the nearest cross-domain instance and then decide whether to approach it. Moreover, with the prototype translation and merging, the nearest cross-domain prototype is more accurate and reliable.

---

> > ### Comment · Reviewer_ssYY · 2024-08-09
> >
> > Thanks for the author's response. I decide to raise my score to Accept.

---

> > > ### Author Response · Authors · 2024-08-09
> > > **Thanks to Reviewer ssYY**
> > >
> > > Thank you for your positive feedback and insightful suggestions. We appreciate your recognition of our efforts to address your concerns and your rating arising. We will be also glad to answer any further questions. Thank you once again for your time and valuable feedback.

---

### Official Review · Reviewer_tjhd · 2024-07-11

**Soundness:** 3
**Presentation:** 3
**Contribution:** 3
**Rating:** 6
**Confidence:** 3

**Summary:**

This paper tackles the problem of unsupervised cross-domain retrieval. This is the problem where the query and retrieval domains are distinct. For example, in sketch to real retrieval, the system must retrieve the most relevant real images to a query sketch. "Unsupervised" refers to the fact that no labels are available during training, but the images from both domains are available. The authors claim to be the first to investigate the "universal" version of this problem, where the query and retrieval domains are allowed to have disjoint labels spaces. For this problem, the authors propose a two-stage optimization procedure. In the first stage, three losses are used: (1) an instance-wise contrastive loss (2) a cluster-wise contrastive loss and (3) a semantic enhanced loss. In the second stage, the embeddings between domains are aligned with three losses: (1) an adversarial domain alignment loss (2) a contrastive loss and (3) a nearest neighbor matching loss.

**Strengths:**

(1) The method is theoretically motivated.

(2) The paper follows a logical orders.

(3) Experiments appear to be complete.

**Weaknesses:**

(1) The method is clearly described and seems to be theoretically motivated. However, it is hard to understand intuitively why each loss is necessary. In particular, why we must use six different versions of the contrastive loss across two stages? (IPM, INCE, PNCE, SEL, SPR, SN2M). The theory only seems to justify the IPM loss.

(2) In my opinion, even for someone well versed in metric learning, this method is hard to grasp. Some examples:

 - In line 148, the method applies k-means with a variable number of clusters determined by the "Elbow approach" and a contrastive loss on top of the cluster centroids. Just this one paragraph requires the person implementing the algorithm to reference another paper and implement a clustering algorithm.

- The argument, starting at line 152, explaining the IPM loss is hard to understand, mostly because of the unusual notation (arrows and xor symbols).

- The argument for the SN2M loss, starting at line 235 is unclear to me.

(3) Overall, the method reads like a series of steps that do not follow one central motivation.

**Questions:**

(1) Why do we need two stages of training? Is it really necessary to have two completely different sets of novel loss functions in each stage?

**Limitations:**

Adequately addressed.

---

> ### Author Rebuttal · Authors · 2024-08-07
>
> > The necessity illustration of each loss and the theory justification
>
> Firstly, we did not use six versions of contrastive loss. **IPM** combines INCE and PNCE with the intuitive goal of performing categorical semantic learning on unlabeled domain data. **INCE** forms the basis of unsupervised semantic learning. However, relying solely on INCE results in unclear boundaries between categories. Fortunately, introducing **PNCE** can enhance the distinction between categories, but determining the optimal timing to introduce PNCE requires careful consideration. We use a sigmoid function to weight the INCE training and then introduce varying degrees of PNCE accordingly. Additionally, we design a prototype conversion strategy to provide a cross-domain unified prototypical structure for PNCE. However, this IPM suffers from single prototype allocation mistakes and errors. To compensate for this issue, we propose **SEL**, which considers the relationship between a single instance and all prototypes, thereby alleviating erroneous optimization caused by single prototype allocation mistakes. **IPM and SEL constitute the first stage of UEM optimization.**
>
> In the second stage of UEM, we recognize that the primary reason for the inaccuracy of existing cross-domain matching work is the neglect of the domain gap. However, regular domain alignment methods are not suitable for our UEM framework as they severely destroy the prototypical structure learned in the first stage. Therefore, we modified standard domain adversarial training by introducing **semantic preservation constraints (SPR)**. We also design a more accurate cross-domain matching algorithm that aligns with the unified prototypical structure, which is **SN^2M**. SN^2M assesses the reliability of cross-domain pairs by the consistency of the nearest prototypes in their respective domains and then decides whether to treat the pair as a positive pair in contrastive learning. Accordingly, **the second stage of UEM optimization is guided by SPR and SN^2M.**
>
> In our current work, we have only conducted theoretical analysis on IPM. Due to the high complexity of entanglement between feature geometry distribution and semantics in high-dimensional space, theoretical analysis of SPR and SN^2M is challenging in supervised learning. Considering unsupervised cross-domain scenarios makes it even more difficult. Thus, we will attempt to provide theoretical explanations for this part in future work.
>
> > Detailed description of clustering and Elbow approach
>
> Thanks for your suggestion. We will incorporate the following description of clustering and the Elbow approach into our future revision. The Elbow approach requires a pre-set maximum cluster number, which is 100 in our implementation. Then we repeatedly apply K-Means to the memory bank of each domain with the cluster number increasing from 2 to 100. For each run, we record two metrics. One is the within-cluster sum of squares (WCSS) which measures the sum of squared distances between each data point and its assigned centroid, reflecting the compactness of the clusters. The other is the silhouette score (SS) which measures how similar a data point is to its own cluster compared to other clusters. Then we draw two curves with the cluster number as the x-axis and the metric value as the y-axis. To determine the elbow point, we select the farthest point below the line passing through the first point and the last point of the curve w.r.t Euclidean distance as the estimated cluster number. After obtaining the respective estimated cluster numbers from the WCSS and SS curves, we select the larger one as the final estimation.
>
>
> > More explanation of the IPM
>
> Please refer to the explanation and description of cross-domain prototype conversion in the Global Response.
>
>
> > More explanation of the SN2M
>
> Please refer to the explanation and description of cross-domain instance matching in the Global Response.
>
> > The central motivation of the two-stage training
>
> As mentioned in the introduction, the successful achievement of cross-domain retrieval (CDR) relies on solving two problems: 1) effectively distinguishing data samples in each domain, and 2) achieving alignment across domains for samples of the same category. Most existing methods adopt a two-stage processing strategy, where self-supervised learning (SSL) is first used for categorical semantic learning on unlabeled data, followed by nearest neighbor searching for cross-domain categorical matching. Our UEM framework also employs this two-stage strategy and introduces several novel designs and algorithms during the training phases to address the U^2CDR problem.
>
> We considered whether it might be possible to solve the U^2CDR problem with a single-stage, end-to-end framework. The answer is no because we cannot introduce additional constraints during the first stage of SSL, such as domain alignment or cross-domain categorical matching. **These constraints would significantly affect the original optimization goals and directions of SSL, deviating the model from learning categorical semantics, which is the basis of everything.** To validate our thinking, we carry out experiments with a single-stage design by incorporating domain alignment (from the second stage of UEM) and SN^2M into the first stage. Below are the experiment results for close-set, partial, and open-set UCDR (in each cell of open-set UCDR, the left number is Shared-set mAP@All, and the right is Open-set detection accuracy). As shown in the table, **the performance of the single-stage process is indeed much poorer compared to the two-stage design.**
>
> |||Office-31|Office-Home|DomainNet|
> |-|-|-|-|-|
> |Close-set UCDR|Single-stage UEM|64.7|40.4|24.9|
> ||UEM|81.9|52.4|34.4|
> |Partial UCDR|Single-stage UEM|45.0|30.5|24.4|
> ||UEM|63.0|46.0|31.3|
> |Open-set UCDR|Single-stage UEM|57.7, 65.3|39.8, 60.9|25.2, 59.8|
> ||UEM|77.4, 92.5|50.2, 86.7|34.8, 80.9|

---

> ### Author Response · Authors · 2024-08-13
> **A Gentle Reminder of Further Feedback to Reviewer tjhd**
>
> Dear Reviewer tjhd,
>
> As the rebuttal discussion phase ends soon, we want to express our gratitude for your engagement thus far. We really want to check with you whether our response addresses your concerns during the author-reviewer discussion phase. We have diligently addressed every concern and question you raised during the initial review, and our efforts are aimed at enhancing the clarity and quality of our work.
>
> We genuinely hope our responses have resolved your concerns related to the design intuition behind our methodology, the detailed description of the Elbow approach, IPM and SN2M losses, and the central motivation of the two-stage design. Your thoughtful evaluation greatly aids in our paper's refinement and strength. Again, we sincerely appreciate your dedication and time.
>
> Best regards,
>
> Authors of Paper 5055

---

> > ### Comment · Reviewer_tjhd · 2024-08-13
> > **Acknowledgement of Rebuttal**
> >
> > Dear Authors,
> >
> > I read the rebuttal and appreciate the additional experiments and clarifications.

---

> > > ### Author Response · Authors · 2024-08-13
> > >
> > > Dear Reviewer tjhd,
> > >
> > > Thank you for your positive feedback and insightful suggestions. We appreciate your recognition of our efforts to address your concerns and include more experiments. We are committed to expanding these clarifications and experiments to our next revised version.
> > >
> > > We value your input and sincerely hope you consider raising your rating based on the improvements we’re implementing. Your endorsement would greatly enhance the credibility of our work. We will be also glad to answer any further questions.
> > >
> > > Best regards,
> > >
> > > Authors of Paper 5055

---

### Official Review · Reviewer_gZ1R · 2024-07-12

**Soundness:** 3
**Presentation:** 2
**Contribution:** 3
**Rating:** 5
**Confidence:** 4

**Summary:**

This paper proposes Universal Unsupervised Cross-Domain Retrieval for the first time and designs a two-stage semantic feature learning framework to address it.

**Strengths:**

This paper proposes a new approach in universal unsupervised domain adaptation, with sufficient experiments to verify its motivation.

**Weaknesses:**

1. In unified unsupervised domain adaptation, there is no handling of instances that are not common categories. Isn't this necessary?

2. From the perspective of innovation, the proposed unified prototype structure is interesting, and the rest is mostly incremental work, such as semantic structure preservation and adjacent feature matching in domain adaptation. From the visualization results, the author failed to prove the above contributions.

3. This paper should reflect the difference between universal domain adaptation and unsupervised domain adaptation.

4. This article does not have a better way to state the method, especially in cross-domain prototype conversion and close neighbor matching.

**Questions:**

See Weaknesses section

---

> ### Author Rebuttal · Authors · 2024-08-07
>
> > Any handling of instances belonging to uncommon categories
>
> In the first stage of our UEM framework, we aim to build a unified prototypical structure across domains via the IPM loss. The IPM loss is a combination of instance and prototype contrastive losses. Given that the IPM loss is computed separately in each domain, we design a novel prototype merging strategy to merge highly potential common categories across domains. Then for the prototype contrastive loss, each instance searches for and tries to approach its closest prototype. **There is no difference between instances corresponding to unmerged (uncommon category) or merged prototypes, as all prototypes should be treated equally to form the final prototypical structure -- otherwise, the uncommon category detection in partial and open-set cross-domain retrieval becomes much harder.** As for the second stage, each instance of each domain searches for and decides whether to approach the nearest neighbor in the other domain. For instances belonging to merged prototypes (common category), there is a much higher possibility of being optimized to approach their nearest neighbors. **While for instances that are not common categories, they usually shouldn't approach their cross-domain neighbors. However, we still allow these instances to approach their domain-translated prototypes, which means that we do have the handling for instances that are not common categories.** We believe this is necessary to achieve more effective cross-domain matching.
>
> > Novelty of semantic structure preservation and adjacent feature matching
>
> Please refer to the Global Response for a detailed description of the novelty. To further validate the novelty of semantic structure preservation and adjacent feature matching, we carry out experiments with the replacement of another state-of-the-art binary graph-based semantic preservation approach (SPPA [1]) and nearest neighbor searching algorithm (CD^2NN [2]). SPPA preserves the instance-level cosine similarity within each cluster and the prototype-level Euclidean distance across clusters. CD^2NN determines cross-domain nearest instance pair by seeking neighboring instance consistency. The experiment results in open-set CDR (in each cell, the left number is the Shared-set mAP@All, while the right is the Open-set detection accuracy) are shown below, **which demonstrate that UEM's semantic structure preservation and adjacent feature matching are more effective.**
>
> |         | Office-31  | Office-Home | DomainNet  |
> |---------|------------|-------------|------------|
> | UEM w/ SPPA  | 76.0, 88.9| 47.2, 84.4  | 31.5, 75.7   |
> | UEM w/ CD^2NN | 76.4, 89.4 |   46.5, 85.0 | 32.1, 76.4 |
> | UEM     | 77.4, 92.5 |  50.2, 86.7  | 34.8, 80.9 |
>
> [1] Lin, et al. Continual semantic segmentation via structure preserving and projected feature alignment. ECCV 2022
>
> [2] Liu, et al. Promoting Semantic Connectivity: Dual Nearest Neighbors Contrastive Learning for Unsupervised Domain Generalization. CVPR 2023
>
> > Difference between universal and unsupervised domain adaptation
>
> We suppose you are indicating the difference between universal and unsupervised cross-domain retrieval (CDR) rather than domain adaptation. First, the CDR problem considers two semantically distinct but similar domains and aims to retrieve data samples from one domain belonging to the same category as a query sample from the other domain. The unsupervised CDR specifies that these two domains only contain unlabeled data. **Regular unsupervised CDR studies assume that the category spaces of these two domains are identical, while universal CDR focuses on scenarios where the category space across domains is distinct.** In our work, we believe the assumption of identical cross-domain category space is unreasonable for many real-world applications, as the categorical composition of an unlabeled data domain is hard to acquire without detailed analysis and dedicated expertise.
>
>
> > Better statement of cross-domain conversion and close neighbor matching
>
> Please refer to the Global Response for better descriptions of cross-domain conversion and close neighbor matching.

---

> ### Author Response · Authors · 2024-08-13
> **A Gentle Reminder of Further Feedback to Reviewer gZ1R**
>
> Dear Reviewer gZ1R,
>
> The conclusion of the discussion period is closing, and we eagerly await your response. We greatly appreciate your time and effort in reviewing this paper and helping us improve it.
>
> Thank you again for the detailed and constructive reviews. We hope our response is able to address your comments related to the handling of uncommon categories, the novelty description of semantic preservation and adjacent feature matching, and clarification of problem settings. We take this as a great opportunity to improve our work and shall be grateful for any additional feedback you could give us.
>
> Best Regards,
>
> Authors of Paper 5055

---

### Author Rebuttal · Authors · 2024-08-07

## Global Response

We would like to thank all the reviewers for their constructive comments and suggestions. In the global response below, we respond to some common questions and present more visualization in the attached PDF.

> [For Reviewers gZ1R and ssYY] Methodology novelty
+ **Unified Prototypical Structure**: We identified the challenges of applying standard contrastive learning (CL) to U^2CDR, with the most significant being the cross-domain geometry distinctness. To tackle this issue, we innovatively perform cross-domain translation on each domain’s prototypes, followed by a unified merging strategy. This ensures a unified prototypical structure during prototype CL. To our knowledge, no prior work has employed such cross-domain prototype conversion as we have.
+ **Single Prototype Allocation Compensation**: In the prototype CL, the single prototype allocation can often be inaccurate, especially in the early stages of training. To compensate for this inaccuracy, we design a semantic-enhanced loss (SEL) that considers the relationship of an instance with all prototypes and directly optimizes the geometric distances between them. We have not seen this specific design in existing research.
+ **Optimal Timing for IPM**: Practically, we need instance CL to gradually learn the semantic prototypical framework of a domain, followed by prototypical CL and SEL to further stabilize and refine this framework. Determining the optimal timing to combine these losses is a challenging issue. For example, UCDIR divides the process into three stages, using different weights to combine instance and prototype CL at each stage. Our UEM resolves this issue by utilizing the Sigmoid function for weighting throughout the training process.
+ **Semantic-preserving Domain Alignment**: We discovered that the primary reason for the inaccuracy of existing cross-domain matching algorithms is the neglect of the domain gap. However, standard domain alignment methods are not suitable for our UEM framework as they severely destroy the prototypical structure learned in the first stage. Therefore, we modified the standard domain adversarial training by introducing semantic preservation constraints. These constraints utilize cosine similarity and Euclidean distance to associate all instances pairwise. This pairwise association is simple yet effective as any change in an instance affects the cosine similarity and Euclidean distance of all other instances. Although geometric structure preservation has been explored in continual learning using multi-node graphs based on graph neural networks [1], our UEM’s semantic preservation approach is much simpler and unseen before.
+ **Mismatch Filtering in Nearest Neighbor Searching**: We found that existing cross-domain retrieval work neglects to handle the inevitable mismatches in nearest neighbor searching, which usually occur at cluster boundaries. Thus, we designed SN^2M to evaluate the consistency of cross-domain neighboring instance pairs relative to their prototypes in each domain and determine whether to optimize the instance pair to be closer. Experimental results show that our SN^2M is highly accurate and perfectly aligns with the previously mentioned prototype translation and merging, distinguishing it completely from existing cross-domain matching algorithms.

[1] Yu, Da, et al. Contrastive correlation preserving replay for online continual learning. IEEE TCSVT 2023

> [For Reviewers gZ1R and tjhd] More explanation about cross-domain prototype conversion and cross-domain instance matching

We elaborate on these two aspects below and will incorporate them in the revision of the paper. To simplify the explanation, we avoid using mathematical symbols here but will do so in paper revision.

+ **Cross-Domain Prototype Conversion**: We design the cross-domain prototype conversion strategy to unify the prototypical structures of different domains. This strategy involves the following four steps:
    + As an example, for prototype conversion of domain A, we first translate all prototypes of domain B along the vector connecting the centers of the two domains to domain A.
    + Next, all prototypes in domain A use the Hungarian algorithm to find the nearest translated domain B prototypes.
    + For each prototype pair determined by the Hungarian algorithm, we check if they satisfy the merging condition (Eq.9 in the paper). If they do, the prototypes are merged by averaging; if not, they remain unchanged.
    + Finally, the prototype set for domain A is composed of the merged prototypes, the unmerged original domain A prototypes, and the unmerged translated domain B prototypes.

We visualize this prototype conversion as Figure 1 in the attached PDF.

+ **Cross-Domain Matching**: Our SN^2M can assess the reliability of cross-domain instance pairs and then decide whether to include them as positive pairs in contrastive learning. The specific steps are as follows:
    + The prototypes of domains A and B are transformed according to the previously described prototype conversion strategy.
    + For an instance $x^A$ in domain A, we find its nearest domain A prototype $p_{x^A}^A$ based on the product of cosine similarity and Euclidean distance. We also find $x^A$'s nearest domain B instance $x_{x^A}^B$ using the same criteria (refer to Eq.15 and Eq.16 in the paper).
    + For the nearest domain B instance $x_{x^A}^B$, we then find its nearest domain B prototype $p^B_{x_{x^A}^B}$.
    + If $p^B_{x_{x^A}^B}$ matches $p_{x^A}^A$ across domains (i.e., if we translate $p_{x^A}^A$ from domain A to B, the translated $\widetilde{p_{x^A}^A}$ is the same as $p^B_{x_{x^A}^B}$), we consider $x^A$ and $x_{x^A}^B$ as a positive pair in contrastive learning (Eq.17). Otherwise, we only consider $x^A$ and $\widetilde{p_{x^A}^A}$ as a positive pair in contrastive learning.

We visualize the SN^2M process as Figure 2 in the attached PDF.

---

### Decision · Program_Chairs · 2024-09-25

**Decision:**

Accept (poster)

**Comment:**

The paper introduces the problem of Universal Unsupervised Cross-Domain Retrieval  for the first time and designs a novel two-stage semantic feature learning framework to effectively address it.  All reviewers have acknowledged the technical novelty and importance of tackling the “universal” problem. The rebuttal effectively addressed most concerns raised by the reviewers, leading to final ratings of Borderline Accept, Weak Accept, and Accept. Overall, the proposed framework offers a robust solution for cross-domain retrieval without supervision. Thus, the AC agrees with the reviewers and recommends "accept".